# The intrinsic time tracker: temporal context is embedded in entorhinal and hippocampal functional connectivity patterns

Jingyi Wang [1], Arielle Tambini [2,3], Laura Pritschet[1,4], Caitlin M. Taylor [1], Emily G. Jacobs[1] & Regina C. Lapate [1] ✉

Changes in task-evoked activity in the entorhinal cortex (EC) and hippocampus have been shown to track changes in temporal context at short and long timescales. However, whether spontaneous changes in EC and hippocampal neural signals—in the absence of task demands—likewise reflect the passage of time remains unknown. Here, we leveraged a dense-sampling study in which two individuals underwent daily resting-state fMRI for 30 days. Similarity in EC- and anterior hippocampal-whole-brain resting connectivity patterns was negatively correlated with the time interval between sessions, suggesting a spontaneous, slow-drifting neural signature of time. These changes could not be explained by other time-varying factors (including session-wise changes in mood, hormones, or motion). Hippocampal connectivity temporal drifts followed an anterior-to-posterior gradient, and anterolateral EC showed stronger temporal drift than posteromedial EC. Finally, posterior networks (including visual and default mode) primarily drove drifts in EC- and hippocampal-whole-brain connectivity over time. Collectively, these findings reveal a resting-state connectivity signature that reflects the passage of time in the absence of task demands and follows a functional gradient along the longitudinal axis of the hippocampus.

*"No man ever steps in the same river twice, for it's not the same river and he's not the same man"* - Heraclitus

In 500 B.C.E., Heraclitus captured humans' elemental sense that time flows continuously, with the world in a state of constant change. How does the brain give rise to our experience of the flow of time? Prior studies suggest that the dynamics of time and temporal context—including temporal intervals, event duration, and the temporal order of events—are reflected in changes in neural activity patterns in the hippocampus (HPC) and its primary source of cortical projections, the entorhinal cortex (EC)[1–14]. For instance, intracranial recordings, calcium imaging, and fMRI recordings in humans and rodents have shown that larger temporal intervals between events are typically associated with greater dissimilarity (i.e., distances) of multivariate neural activity

patterns in both the HPC[13,15–29] and EC[30,31]. Critically, in the rodent HPC, cell ensemble firing patterns gradually change over time even when other features in the environment (e.g., spatial features) remain unchanged[16,17,19,27,32].

Prior studies indicate that these putative neural signatures of temporal context extend beyond the length of a single experimental session and are evident over timescales of days and months[33]. For example, the similarity structure of neural activity patterns of time cell ensembles in the mouse HPC can be used to decode distinct recording sessions over a four-day period[27]. In a naturalistic experiment, human participants viewed pictures of self-relevant events spanning a one-month period in the MRI scanner[23]. The objective time interval elapsed between those events correlated with greater dissimilarity of

[1]Department of Psychological & Brain Sciences, University of California, Santa Barbara, Santa Barbara, CA, USA. [2]Center for Biomedical Imaging and Neuromodulation, Nathan S. Kline Institute for Psychiatric Research, New York, NY, USA. [3]Department of Psychiatry, New York University Grossman School of Medicine, New York, NY, USA. [4]Department of Psychiatry, University of Pennsylvania, Pennsylvania, PA, USA. ✉e-mail: lapate@ucsb.edu

multivariate neural activity patterns in the anterior HPC[23]. Further, a recent dense-sampling fMRI study showed that the similarity of stimulus-evoked multivariate neural activity patterns in the HPC and EC tracked temporal contexts over a 10-month period[34]. Collectively, these results lend credence to the river metaphor, indicating that both the HPC and EC may carry representations of time or temporal context across long timescales when assessed in a stimulus-evoked fashion. However, these findings raise an intriguing but largely unanswered question: Do intrinsic, long-timescale fluctuations in HPC and EC signals spontaneously track or reflect the passage of time—i.e., in task-free contexts? In the current study, we used resting-state human fMRI data to answer this question.

The HPC can be divided into anterior and posterior portions (corresponding to ventral and dorsal HPC in rodents)[35,36], which are differentially connected to the EC[37–39]. Extant data suggest that temporal context representations may be differentially supported by anterior *versus* posterior HPC[23] as well as by lateral *versus* medial EC subregions[31,40,41]. Previous results suggest a prominent role for anterior (vs. posterior) HPC in tracking temporal information over long timescales (e.g., over a day[42] and a one-month period[23]). This is consistent with prior work indicating that the anterior HPC (aHPC) represents information with a relatively coarser coding scheme, including in space (e.g., aHPC/ventral HPC receptive fields are larger than those in posterior HPC (pHPC)/dorsal HPC)[43,44], temporal receptive windows (e.g., a slower temporal autocorrelation decay has been noted in aHPC than pHPC)[45–47], and in episodic memory (e.g., 'gist-like' memory has been linked to aHPC function, whereas detailed autobiographical memories to pHPC)[35,36,48,49]. This putative functional gradient in coding scheme along the longitudinal axis of the HPC may also extend to coding of temporal context[50]. Specifically, using a time interval estimation task, Polti and colleagues found higher aHPC activity in trials with time estimates towards the mean of recently-sampled temporal intervals, whereas pHPC activity more closely reflected the time interval of the current trial, suggesting that aHPC signals incorporated longer timescales than signals in pHPC[51]. Across a number of studies examining temporal memory in naturalistic settings—with varied task manipulations including virtual environment navigation[21], screenshots from life simulation computer games[52], and complex narratives presented in audio clips[30]—only the aHPC (not pHPC) tracked subjectively-remembered time. Relatedly, evidence for task-evoked temporal context representations has typically been stronger in the human anterolateral division of EC (alEC, analogous to the lateral division in rodents) than in the posteromedial division (pmEC, analogous to the medial division in rodents)[10,31,33,40,41,53]. Collectively, these results obtained in task contexts suggest that coding of temporal contexts at longer timescales may be differentially supported across distinct regions of the HPC (i.e, in aHPC *versus* pHPC) and EC (in alEC *versus* pmEC). However, whether the strength of spontaneous time-varying representations in humans varies systematically between EC subregions or along the HPC longitudinal axis—putatively reflecting an anterior-posterior gradient of temporal context coding—has not been previously tested.

Previous tracing[37–39,54] and functional connectivity studies[55–57] indicate that EC and HPC subregions also show differential inter-regional and cortical connectivity profiles. For instance, the aHPC and medial EC exhibit preferential connectivity with the default mode network (DMN), whereas the pHPC and lateral EC are strongly connected with the ventral attention network (VAN/salience network)[53,55–58]. Given previous findings relating EC and HPC function to temporal-context-relevant representations[5,33], it is possible that their system-level interactions may also be sensitive to the passage of time.

Here, we investigated whether multivariate EC and HPC signals intrinsically reflect the passage of time in humans in a task-free context (i.e., during rest) over a relatively long timescale of 30 days. In addition, we tested whether the magnitude of time-dependent changes systematically varied along the HPC longitudinal axis or differed between EC subregions. To do so, we leveraged a dense sampling study in which two participants underwent repeated resting-state fMRI scans at fixed times each day for 30 consecutive days (Fig. 1)[59–62]. Going beyond intra-regional metrics, we took the approach of examining HPC and EC low-frequency (LF) fluctuations in the BOLD signal as reflected by changes in HPC- and EC-whole-brain resting state functional connectivity (henceforth, resting connectivity) patterns. Large-scale whole-brain connectivity pattern changes have been previously shown to track slow changes in emotion[63] and other behavioral states[64–67]. Thus, using this approach, we tested whether spontaneous changes in EC- and HPC-whole-brain resting connectivity patterns tracked the objective passage of time over a month in a task-free, resting state.

We hypothesized that the similarity of EC- and HPC-whole-brain resting connectivity patterns between sessions would negatively correlate with objectively elapsed time between sessions[4,6,9,33]. We further hypothesized that the strength of time-related changes in HPC would systematically vary along the longitudinal axis, with stronger temporal drifts in anterior than posterior HPC over a one-month period. Moreover, we predicted that time-related drifts would be particularly reliable in alEC compared to pmEC. We controlled for several potentially confounding time-varying factors, including inter-session fluctuations in mood, motion, and hormones. To foreshadow, we found that changes in intrinsic EC- and HPC-whole-brain resting connectivity patterns reflected the passage of time in a task-free context in humans, with the strength of hippocampal time-related drifts varying systematically along the longitudinal axis of the human HPC.

## Results
### EC- and HPC-whole-brain resting connectivity patterns reflect elapsed time
Given that HPC and EC signals have been found to track temporal information in structured, task-dependent contexts[1–5,8,9], we investigated whether their large-scale resting connectivity patterns systematically drift with elapsed time. To examine whether elapsed time was reflected in the similarity of large-scale resting connectivity patterns, we first obtained patterns of whole-brain resting functional connectivity for each session using EC and HPC (including aHPC and pHPC) as seed regions of interest (ROIs). Next, we computed the pattern similarity between each pair of sessions for each ROI. This allowed us to calculate a 'temporal drift score', or the correlation between pattern similarity and time interval across session pairs (Fig. 1C−E, see Methods for details). As mentioned, we hypothesized that whole-brain resting connectivity pattern of both ROIs would become less similar (i.e., negatively correlate) with the time elapsed between sessions[4,6,9,33].

Accordingly, EC- and HPC-whole-brain resting pattern similarities correlated negatively with the time interval elapsed between session pairs for both female (EC: $r = -0.206$, $p < 0.001$; HPC: $r = -0.187$, $p < 0.001$) and male (EC: $r = -0.217$, $p < 0.001$; HPC: $r = -0.146$, $p < 0.001$) subjects (all nonparametric $p$-values were derived from a null distribution obtained by shuffling time interval labels for session pairs ($n = 5000$ shuffles); Fig. 2, Supplementary Table 1). Of note, aHPC-whole-brain resting connectivity patterns showed stronger temporal drift (as indexed by a more negative drift score) than pHPC in both subjects (Female aHPC: $r = -0.180$, $p < 0.001$ vs. pHPC: $r = -0.142$, $p = 0.002$; Male aHPC: $r = -0.153$, $p < 0.001$ vs. pHPC: $r = -0.123$, $p < 0.001$), suggesting a possible functional differentiation of temporal context coding along the longitudinal axis of the HPC (see section below). As expected, the similarity of EC-HPC resting connectivity patterns also decreased over time (Supplementary Results). Collectively, these results indicate that EC- and HPC-whole-brain resting connectivity patterns become increasingly dissimilar with longer time intervals, suggesting that large-scale EC and HPC signals in humans intrinsically reflect the passage of time.

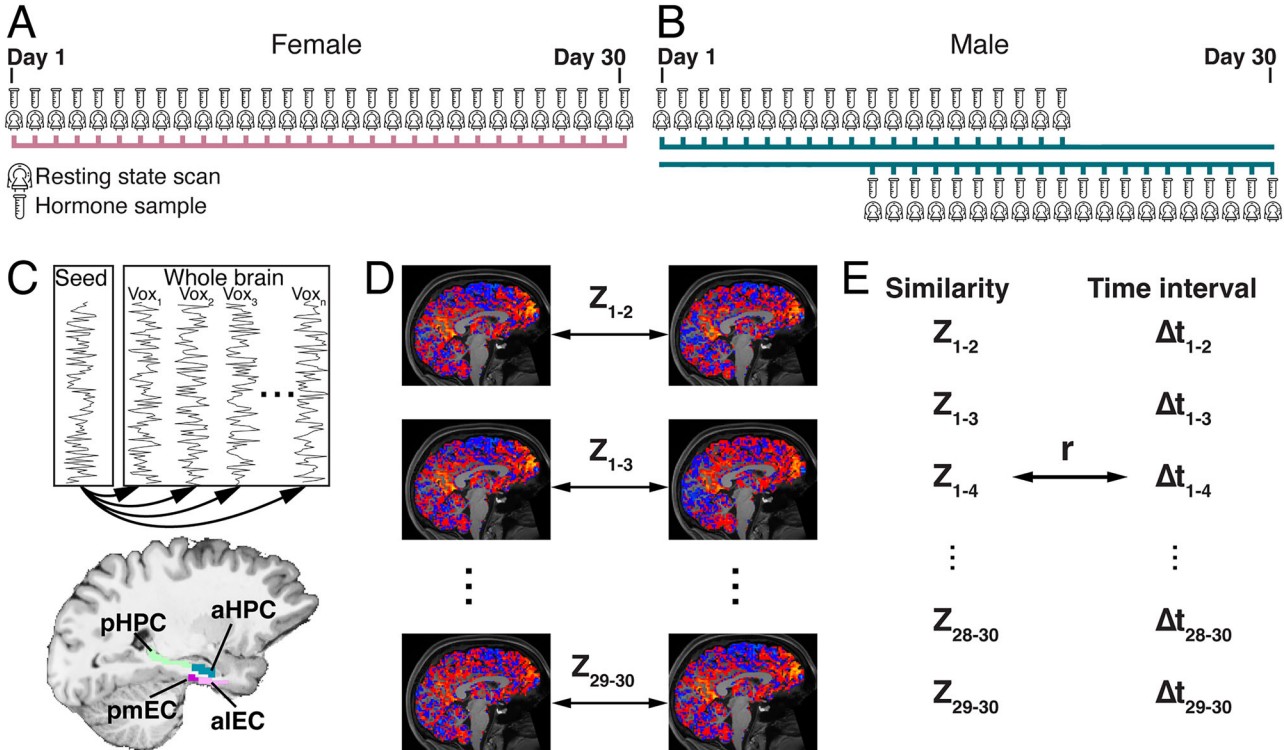

**Fig. 1 | Experimental procedure and calculation of temporal drift scores.**
Sampling rate and study timeline are shown over a 30-day period for the two participants (**A:** Female, **B:** Male). In each session, a resting-state scan was collected. Hormone sampling preceded each scan. Method for calculating temporal drift scores. **C** For each ROI (region of interest, here, the seed region), the resting-state functional connectivity pattern for each session was computed (i.e., the correlation between the seed ROI time series and each gray matter voxel across the whole brain). Then, **D** the similarity between resting connectivity patterns was measured for all pairs of sessions (e.g., $Z_{1-2}$ corresponds to the similarity (Fisher's Z-transformed correlation) between sessions 1 and 2). **E** For each ROI, we obtained a temporal drift score (correlation coefficient), which reflects the association between resting-state pattern similarity and the time interval between sessions.

## Time-related changes in HPC-whole-brain connectivity patterns show an anterior-to-posterior gradient

Given previous evidence suggesting that the aHPC may play a more prominent role than pHPC in tracking time at long timescales in humans[23,35,36,50], we hypothesized that the magnitude of time-related changes in hippocampal connectivity patterns would vary systematically along the longitudinal axis of the HPC. To quantify time-dependent associations along the longitudinal axis of the HPC and examine whether the strength of the association between the similarity of HPC-whole-brain resting connectivity patterns and time varied along the longitudinal axis, we computed temporal drift scores along the HPC on a voxel-wise basis. We found that the aHPC showed numerically stronger time-related resting connectivity pattern drifts than pHPC (Fig. 3A, D), as indexed by a more negative correlation coefficient in aHPC voxels than pHPC voxels for both subjects (Fig. 3B, E). Next, we investigated whether there was a linear association between the magnitude of time-dependent drifts and the longitudinal HPC axis by correlating hippocampal voxel-wise temporal drift scores with their y-axis coordinate (see Methods for details). We found that temporal drift scores were negatively correlated with the anterior-posterior hippocampal position (y-axis coordinates) (Female: Pearson's correlation $r = -0.099$; $p = 0.01$, Male: $r = -0.160$; $p < 0.001$, Supplementary Fig. 1), indicating that they systematically varied along the anterior-to-posterior axis, in a robust association that was significantly above chance (non-parametric permutation test (y-axis coordinate shuffle; $n = 5000$): Female: $p = 0.004$, Male: $p < 0.001$, Fig. 3C, F). Collectively, these results indicate that the strength of time-related changes in HPC-whole-brain resting connectivity patterns varies systematically along the hippocampal longitudinal axis.

## Stronger time-correlated changes in alEC than pmEC resting connectivity patterns

EC is a complex structure that can be subdivided into the alEC and pmEC, which are associated with functionally distinct cortical networks[55,58] and have been differentially linked to temporal coding[40,41]. Here, we found that only alEC-whole-brain resting pattern similarity reliably correlated with the time interval elapsed between session pairs (Female: $r = -0.218$, $p < 0.001$; Male: $-0.179$, $p < 0.001$), whereas the pmEC did not show a consistent association across subjects (only in the Male subject: $r = -0.111$, $p = 0.001$; Female: $r = -0.034$, $p = 0.478$; Fig. 4 A & D, Supplementary Table 1). Importantly, temporal drift scores were significantly more negative in alEC compared to pmEC in both subjects (Female: $z = -3.138$, $p = 0.001$; Male: $z = -1.802$, $p = 0.036$).

To further examine potential functional differences in temporal tracking between the alEC and pmEC, we calculated voxel-wise temporal drift scores and averaged them per subregion. We found that the averaged time-related drift in alEC-whole-brain resting state functional connectivity was consistently stronger than that of the pmEC, as shown by numerically more negative temporal drift scores in the alEC across both subjects (Fig. 4B, C and E, F). Collectively, these findings indicate that spontaneous changes in alEC-whole-brain connectivity correlate with objective time changes.

## Comparing the strength of time-related drifts in EC and HPC *versus* control regions

Next, we tested whether time-related drifts in whole-brain resting connectivity patterns are generally present when using other regions as seeds, *versus* whether they are statistically stronger in EC- and aHPC-

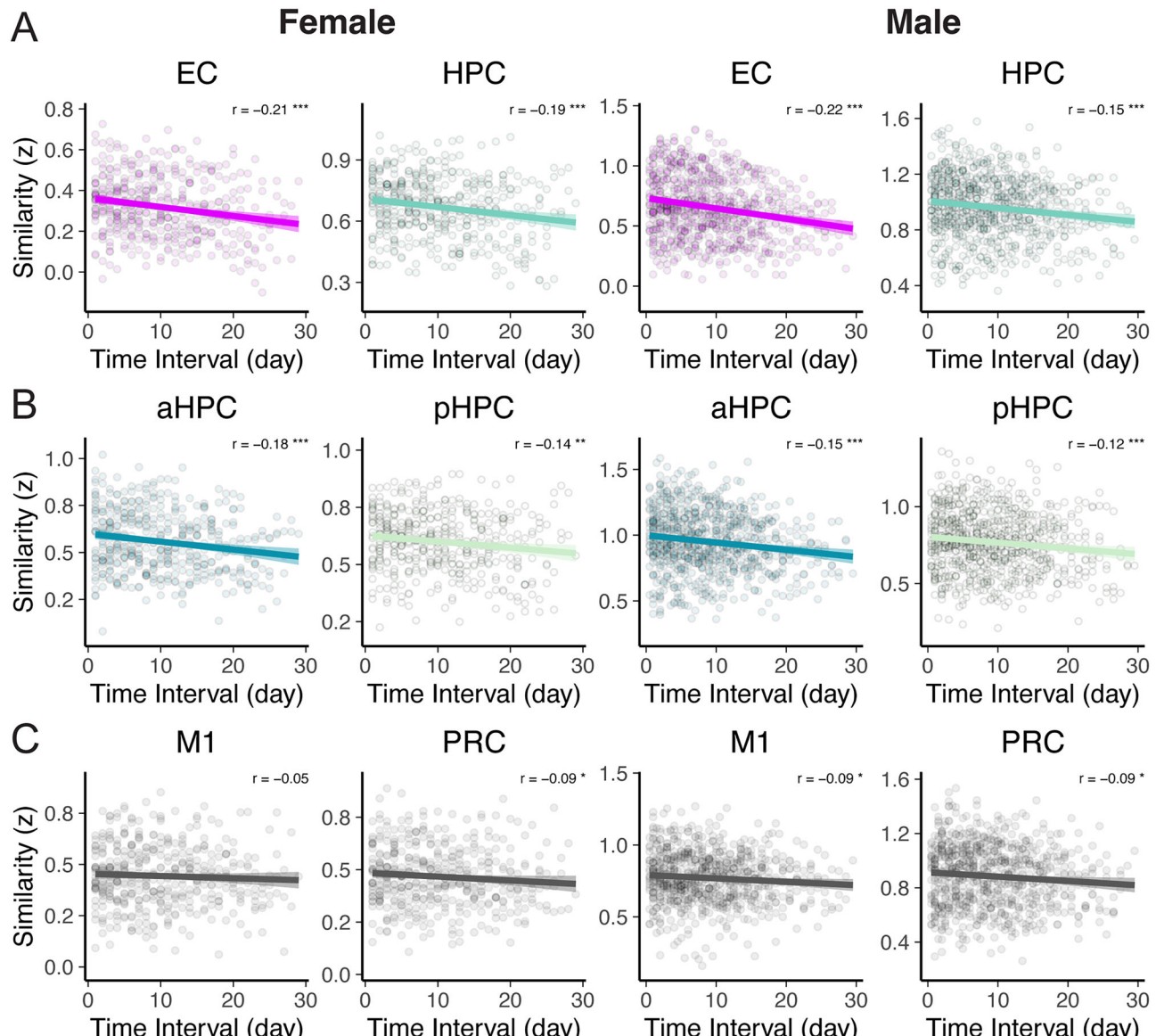

**Fig. 2 | EC- and HPC-whole-brain resting connectivity patterns drift over time.**
The similarity of EC- and HPC-whole-brain resting connectivity patterns decreases (i.e., negatively correlates) with the time interval elapsed between MRI sessions in both subjects. **A** EC and HPC, **B** aHPC and pHPC, and **C** control sites: M1 and PRC. Left panels: female subject, right panels: male subject. Temporal drift scores (Pearson's *r* values) are shown on the top right in each figure. All *p* values are derived from a null distribution by shuffling time interval labels for session pairs ($n = 5000$ shuffles). * $p_{vs. chance} \leq 0.05$; ** $p_{vs. chance} \leq 0.005$; *** $p_{vs. chance} \leq 0.001$. EC entorhinal cortex, HPC hippocampus, aHPC anterior hippocampus, pHPC posterior hippocampus, M1 primary motor cortex, PRC perirhinal cortex. Shaded ribbon: 95% confidence interval (CI).

resting connectivity patterns compared to other regions. To that end, we compared the strength of their time correlation against two control regions: the primary motor cortex (M1) and the perirhinal cortex (PRC). M1 was tested as a control site because contextual representations that track elapsed time are not known to be reliably present in this region[68,69]. The PRC was chosen due to its spatial and anatomical proximity to our other a-priori medial temporal lobe (MTL) ROIs and its established role in primarily coding for item-centered (vs. temporal-context) information in the service of memory, including object-related and semantic information[18,34].

We found that temporal drift scores in the EC—particularly alEC—and in the aHPC were stronger than in both control ROIs (i.e., significantly more negative; test of the difference between dependent correlation coefficients: Female EC vs. control ROIs *ps* < 0.016; Female aHPC vs. control ROIs *ps* < 0.043; Male EC vs. control ROIs *ps* < 0.004; Male aHPC vs. control ROIs *ps* < 0.05, Female alEC vs. control ROIs

*ps* < 0.006; Male alEC vs. control ROIs *ps* < 0.027; Supplementary Table 1). Of note, this was not the case for the whole HPC (which showed significantly stronger drifts than the control ROIs in the Female subject only), pHPC, or pmEC ROIs (which did not differ significantly from control ROIs in either subject). In sum, these results suggest regional specificity in the strength of time-related EC- and aHPC-whole-brain resting connectivity changes.

## Controlling for time-varying factors
To ascertain process specificity of the association between changes in neural activity patterns and temporal context, it is important to isolate intrinsic time from other time-varying factors. While our current design allowed us to hold constant a number of potentially confounding experimental factors across the 30-day period—such as spatial location, equipment, and experimenter—other time-varying factors may still influence resting connectivity patterns, including, for

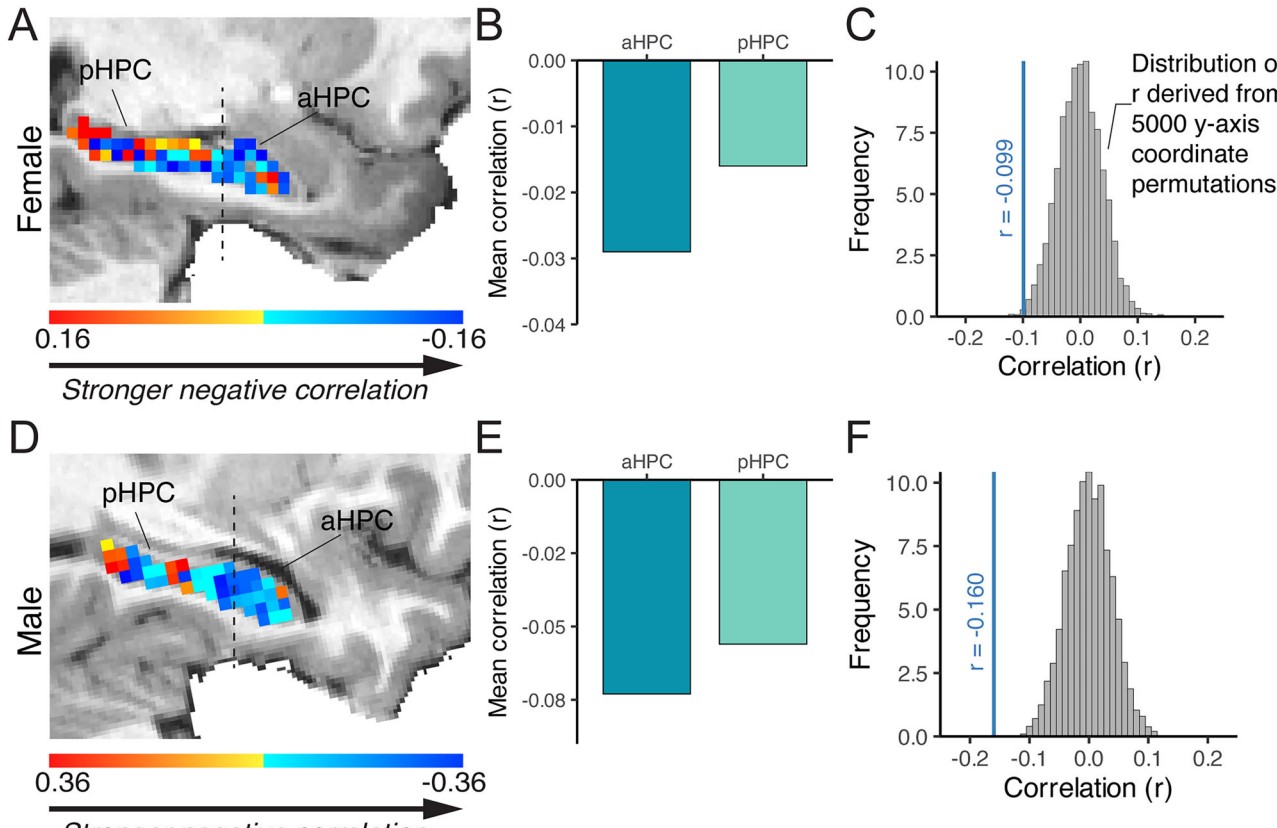

**Fig. 3 | A gradient of strength of temporal drifting along the hippocampal longitudinal axis.** Temporal drift scores are shown for each voxel in the HPC of the female (**A**) and male (**D**) subjects. Temporal drift scores for aHPC and pHPC voxels are shown, indicating a more negative association with time in aHPC than pHPC in both female (**B**) and male (**E**) subjects. Non-parametric permutation tests show the above-chance associations between HPC temporal drift scores and the HPC y-axis coordinates for the female (**C**) and male (**F**) subject. The blue vertical lines show the correlation between voxel-wise temporal drift scores and the HPC y-axis coordinates, which were significantly above chance for both subjects (*n* = 5000 permutations).

instance, head motion differences[70,71], emotional state changes[72], and hormonal fluctuations[59,73]. Therefore, to control for these factors in our study, we extended the regression model used to derive temporal drift scores into a multiple regression analysis framework, which allowed us to control for multiple variables (covariates). These covariates included motion, emotional state changes, as well as hormone fluctuations (Female: estradiol, progesterone, LH and FSH, Male: estradiol, testosterone, cortisol) (see Methods for details). We found that the time interval between sessions significantly predicted the similarity of EC and aHPC resting connectivity patterns after controlling for changes in motion, emotion, and hormones in both participants (Female: EC: B = −0.004 (SE = 0.001), *t* = −4.037, *p* < 0.001; aHPC: B = −0.005 (SE = 0.001), *t* = −3.813, *p* < 0.001; Male: EC: B = −0.016 (SE = 0.003), *t* = −4.820, *p* < 0.001; aHPC: B = −0.008 (SE = 0.003), *t* = −2.687, *p* = 0.008) (Supplementary Fig. 2). Conversely, after controlling for these factors, the similarity of whole-brain resting patterns in the control ROIs no longer correlated with the time interval between sessions (Female: M1: B = −0.0003 (SE = 0.001), *t* = 0.313, *p* = 0.754; PRC: B = −0.004 (SE = 0.001), *t* = −0.446, *p* = 0.656; Male: M1: B = 0.0002 (SE = 0.002), *t* = 0.09, *p* = 0.928; PRC: B = −0.001 (SE = 0.002), *t* = −0.411, *p* = 0.681). Instead, hormonal fluctuations remained the only significant predictor in both multiple regression models for the control ROIs (Supplementary Fig. 3), consistent with prior findings showing the robust influence of hormonal changes on resting-state connectivity throughout the brain[59,61]. Collectively, these results indicate that changes in EC- and aHPC-whole-brain resting functional connectivity patterns correlate with the passage of time beyond changes in neural similarity driven by other time-varying factors.

## EC and aHPC time-related pattern changes were driven by specific cortical networks

Prior work indicates that the DMN regions that are anatomically closest to the MTL are involved in spatiotemporal processes that support episodic memory[74]. This portion of the DMN—comprising the MTL cortex, retrosplenial cortex, posterior cingulate cortex, and the posterior portion of the inferior parietal cortex—is often termed the 'posterior-medial network'[75,76], and corresponds to the DMN-C (also referred to as the DMN-medial temporal lobe subsystem (DMN-MTL)[77] or DMN-A[57,58,78–80]) of a widely used network parcellation (Yeo-17)[74,81–83]. In addition, task-evoked representational activity patterns in the visual cortex have been found to relate to coding of temporal context[84], suggesting that time-dependent connectivity pattern changes in the EC and aHPC may be particularly pronounced when examining interactions with the default mode and visual networks. Of note, EC and aHPC temporal drift scores showed substantial variance across the 17-Yeo networks[83] (non-parametric Kruskal-Wallis test (*n* = 5000 permutations) aHPC: H = 22.901, *p* = 0.029 and EC: H = 19.311, *p* = 0.088) (see Methods for details). Therefore, we next explored whether time-related drifts in EC and aHPC resting connectivity patterns were driven by specific large-scale cortical networks. We compared the putative contributions of large-scale cortical networks relative to the somato-motor network, an a priori control network in which we did not expect to find time-varying representations[68,69].

We found that EC time-related connectivity pattern changes were primarily driven by the DMN-C (Female: r = −0.206, $p_{FDR}$ < 0.001; Male: r = −0.252, $p_{FDR}$ < 0.001) and DMN-D (also known as the temporal-parietal network[85–87] or the auditory/language network[88]; Female:

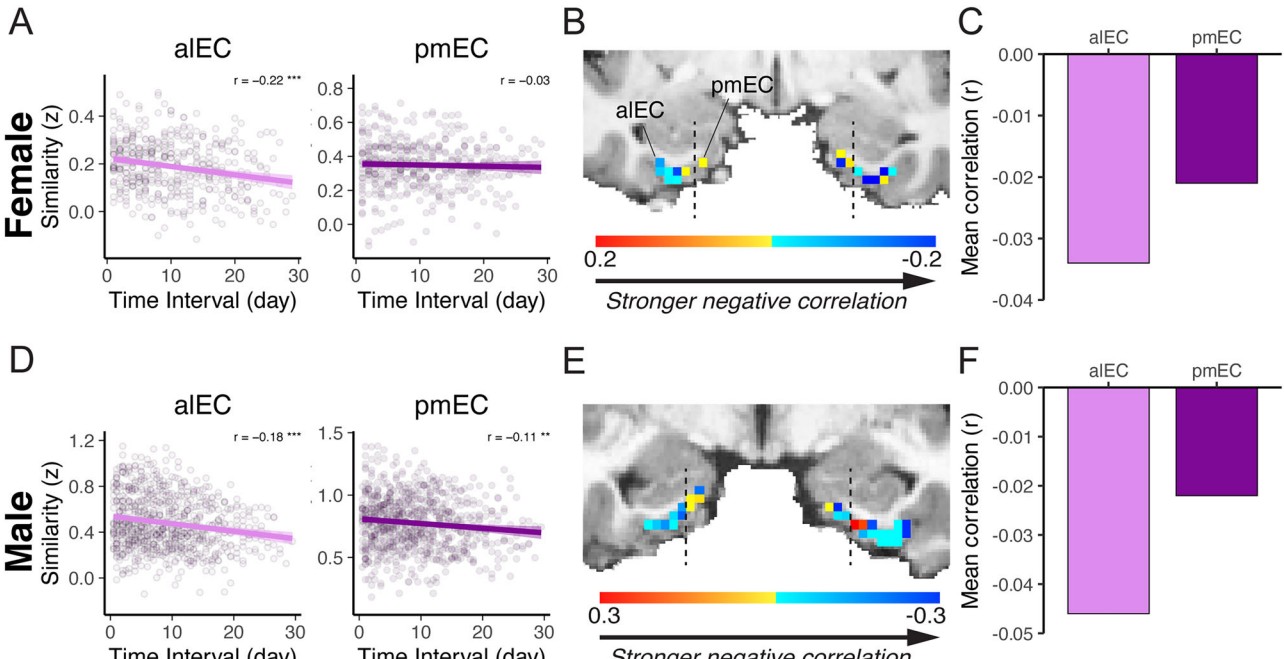

**Fig. 4 | Temporal drift strength differs between EC subregions. A, D** In both subjects, the similarity of alEC-whole-brain resting connectivity patterns decreases with time interval elapsed between sessions, an association not as reliable (across both subjects) in pmEC. Voxel-wise temporal drift scores in EC for the female (**B**) and male (**E**) subjects. Averaged temporal drift scores across alEC and pmEC voxels, demonstrating a stronger negative association with time in alEC compared to pmEC for both female (**C**) and male (**F**) subjects. Shaded ribbon in (**A, D**): 95% CI.

r = −0.175, $p_{FDR}$ = 0.002; Male: r = −0.193, $p_{FDR}$ < 0.001) as well as the dorsal attention network A (DA-A) (Female: r = −0.238, $p_{FDR}$ < 0.001; Male r = −0.275, $p_{FDR}$ < 0.001) and ventral attention network A (VAN-A) (Female: r = −0.188, $p_{FDR}$ < 0.001; Male r = −0.173, $p_{FDR}$ < 0.001, Fig. 5A). These networks showed reliable temporal drifts, which were significantly stronger than the somatomotor network in both subjects ($ps$ ≤ 0.05; Fig. 5C, See Supplementary Tables 3). Similarly, aHPC time-related connectivity pattern changes were driven primarily by the VAN-A (Female: r = −0.253, $p_{FDR}$ < 0.001; Male: r = -0.223, $p_{FDR}$ < 0.001) and the visual network (Visual-A) (Female: r = −0.245, $p_{FDR}$ < 0.001; Male: r = −0.290, $p_{FDR}$ < 0.001, Fig. 5B), where temporal drift scores were reliable (and significantly stronger than in the somatomotor network) in both subjects ($ps$ ≤ 0.043; Fig. 5D, See Supplementary Tables 4). Of note, these results were largely replicated when using an individualized network parcellation method[89,90] (for details, see Supplementary Results: Individualized network parcellation analysis).

## Discussion

Extant work has shown that signals in the EC and HPC reflect changes in temporal context across species and a variety of task demands[1,3–6,8,33]. Here, we investigated whether intrinsic fluctuations in EC- and HPC-whole-brain functional connectivity patterns reflect the passage of time in the absence of task demands. Using a dense-sampling study, we found that both EC- and HPC-whole-brain resting connectivity patterns became increasingly dissimilar with longer time intervals across a 30-day period. Moreover, time-related changes in the HPC showed a functional gradient along the longitudinal axis, wherein aHPC was characterized by stronger time-dependent drifts compared to pHPC. Likewise, time-related drifts were stronger in alEC than pmEC. Critically, time-related changes in whole-brain connectivity patterns were significantly stronger in the EC and aHPC when compared with control regions (motor cortex and nearby MTL cortex), and remained significant after controlling for time-varying factors that fluctuated over the 30-day period, such as hormonal and emotional-state changes. Finally, we found that time-related drifts in

EC and HPC connectivity patterns were not uniform across the brain, but primarily driven by functional connectivity with specific cortical networks (including default mode and visual networks). Collectively, these data suggest that EC- and aHPC-whole-brain resting functional connectivity patterns spontaneously reflect the passage of time in humans.

## EC- and aHPC-whole-brain connectivity patterns reflect the passage of time

A recent surge of empirical work across species has consistently implicated EC and HPC function in temporal context and temporal memory coding[1,3–6,8,9,33,91]. Specifically, the similarity of multivariate neural activity patterns in EC and HPC evoked by stimuli such as images, audio clips, or objects in a virtual environment progressively decreases for stimuli presented further apart in time[12,18,21,23,24,28,30,31], in agreement with the time-related changes in the similarity of global EC and HPC resting connectivity patterns observed here. Of note, changes in multivariate pattern similarity in these regions have been shown to correlate not only with 'objective' variations in time or temporal context at encoding, but also with subjectively remembered temporal intervals and durations[30,92–94], suggesting that time-related changes in neural pattern similarity may sculpt temporal memory.

Going beyond task- or stimulus-evoked neural activity patterns, our study examined low-frequency fluctuations in resting state BOLD fMRI signals to uncover a novel, spontaneous temporal neural signature in EC and aHPC operating at a relatively long timescale (30 days). Large-scale functional-connectivity patterns have previously been shown to track mental states[63–66,95], with the specific nature of mental states that is captured varying as a function of which brain region is used to derive functional connectivity patterns (e.g., amygdala vs. visual cortex). For instance, changes in amygdala-whole-brain functional connectivity patterns have been shown to track emotional states[63], whereas visual cortical connectivity differences predict attentional states during visual processing[65,95]. As HPC and EC play crucial roles in temporal coding and memory, we reasoned that

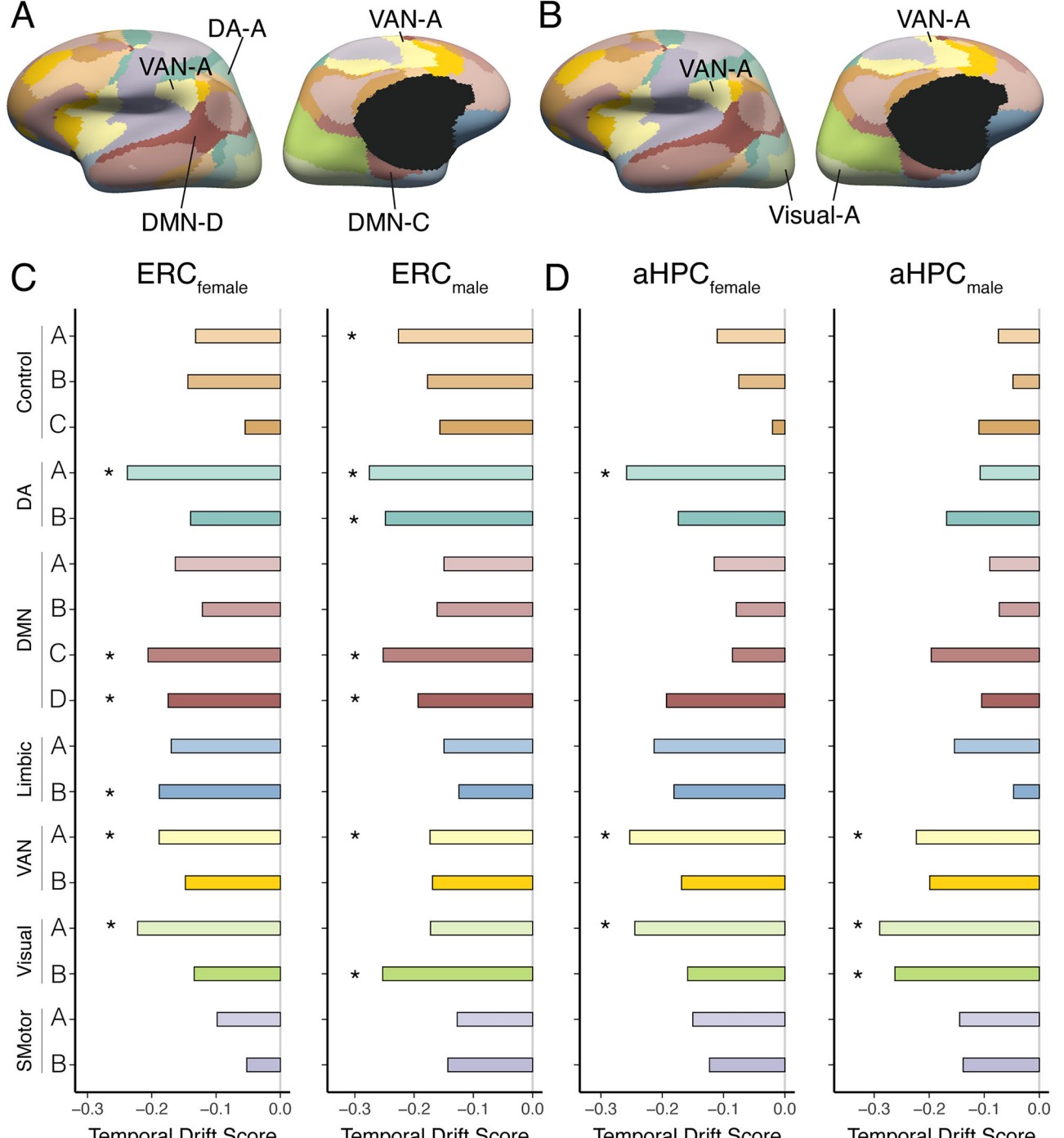

**Fig. 5 | EC and aHPC time-dependent pattern changes were driven by specific networks. A, B** The Yeo-17 networks are shown on a surface template (color-coded by network). The primary networks that drove time-related resting connectivity pattern changes in the **A** EC and **B** aHPC are highlighted. Bar plots show temporal drift scores for **C** EC and Yeo-17 large-scale networks (Female: left, Male: right) and **D** aHPC and Yeo-17 large-scale networks (Female: left, Male: right). *Denotes significantly different temporal drift scores compared to the control somatomotor network in each subject. DMN default mode network, dorsal attention network (DA), ventral attention network (VAN), somatomotor network (SMotor).

changes in HPC- and EC-whole-brain connectivity patterns may track or reflect changes in temporal context.

Accordingly, we found that the gradual drift of HPC- and EC-whole-brain functional connectivity patterns correlated with time elapsed over a 30-day period in experiments devoid of a task, i.e., in spontaneous activity during rest. Importantly, the negative association between elapsed time and whole-brain EC and aHPC connectivity pattern similarity remained significant after controlling for time-

varying factors known to influence the BOLD signal at rest, including emotion dynamics[72], hormonal fluctuations[59], and differences in head motion[70,71]. Of note, several other potentially confounding factors were kept constant across the 30-day period in our study, including spatial location, experimenter, and scanning procedures. Collectively, these findings suggest that changes in the similarity of EC- and aHPC-whole-brain functional connectivity patterns spontaneously reflect the passage of time. Moving forward, it will be important to determine

whether time-related changes observed in EC- and HPC-whole-brain connectivity patterns serve a functional role in episodic memory, putatively providing 'time stamps' that may bind to (and be subsequently accessed with) specific experiences[34,96].

### Functional gradient of time-related connectivity pattern changes along the hippocampal longitudinal axis

Our finding that aHPC-whole-brain resting connectivity patterns showed stronger time-dependent drifts compared to pHPC aligns with prior human work, which has more often uncovered temporal context coding in aHPC than in pHPC[21,23,28,30,52,97]. For example, searchlight analyses have more often identified clusters of voxels within the aHPC (pHPC) that track or represent time intervals, conjunctive spatiotemporal information, and the temporal structure of event sequences[21,28,52,97]. Similarly, in response to images shown twice over a 28-day interval, neural activity patterns in the aHPC showed reduced similarity compared to when images were repeated over a shorter interval, with no such changes observed in the pHPC[98,99]. In a study examining changes in task-evoked hippocampal activity patterns over a similar time scale as our current study, Nielson and colleagues (2015) found that larger temporal distances between specific autobiographical memories were associated with increasingly dissimilar neural activity patterns in the aHPC (but not pHPC) over a one-month period. Moreover, a relatively more rapid decline in the rate of temporal autocorrelation has been noted in the pHPC (vs. aHPC), suggesting that its neural activity patterns may more quickly revert to previous states, potentially limiting systematic time-correlated drift over extended time periods[46,47]. Collectively, these studies converge to suggest the aHPC may be sensitive to changes in time (or temporal contexts) over long timescales in humans—to a greater extent than the pHPC. Our finding that the strength of time-related drift in HPC signals covaried linearly with the position along the HPC longitudinal axis provides additional evidence for a potential anterior-to-posterior functional gradient of time coding in the HPC.

### Time-related whole-brain connectivity pattern changes are stronger in alEC than pmEC

Recent electrophysiological studies in rodents[10,41] and fMRI studies in humans[31,40] indicate that the alEC reliably tracks temporal context, whereas the pmEC does so less consistently. For instance, neurons in the rat lateral EC robustly encode temporal information across timescales from seconds to hours, a phenomenon less pronounced in the medial EC[41]. Similarly, increased BOLD activity in the human alEC—but not in pmEC—was associated with greater accuracy in retrieving temporal information about when an event occurred[40]. In line with these findings, we found that alEC-whole-brain resting connectivity patterns exhibited a stronger and more reliable temporal drift across subjects compared to pmEC, suggesting that spontaneous changes in alEC-whole-brain connectivity patterns may reflect the passage of time.

### Posterior networks drive time-related EC and aHPC neural drifts

Here, we found that time-dependent changes in the similarity of EC- and aHPC- connectivity patterns were driven by specific cortical networks. Specifically, the resting connectivity between EC and the DMN-C showed particularly strong time-dependent changes, which may be related to this network's role in representing contextual features that are critical for mnemonic processes. Stronger co-activation of DMN-C nodes has been found during the retrieval of memory details as well as episodic memory construction[77,100-102]. Moreover, the strength of interconnectivity between DMN-C nodes has been implicated in episodic memory quality. For instance, the strength of resting functional connectivity between DMN-C nodes has been found to positively correlate with the ability to recall spatiotemporal event information[103], and the integrity of the cingulum bundle that links nodes of DMN-C has been found to positively correlate with the number of event element

details (e.g., spatiotemporal context) during free-recall[104]. Given that the EC is part of the DMN-C[83], time-dependent EC-DMN-C resting connectivity pattern drifts may contribute to representing temporal context information that can be incorporated into episodic memory processes.

The visual network also contributed to time-related changes in EC and aHPC global resting connectivity patterns in our study. Multi-session calcium imaging in rodents and fMRI studies in humans have uncovered neural activity drifts at the population level in response to identical visual stimuli shown over a multi-week period[105-107]. Moreover, a recent human fMRI study showed that neural activity patterns in lateral occipital cortex are sensitive to temporal context changes at encoding, suggesting that the visual system may encode temporal information[84], and that temporal-context relevant signals that can be integrated into memory may be available at early stages of sensory encoding.

Finally, we found evidence that time-dependent changes in the connectivity of the EC with the DA-A and DMN-D networks, as well as between the aHPC and the VAN-A network, contributed to large-scale time-related drifts uncovered in our study. Previous studies underscore the role of the VAN and DA networks in allocating limited attentional resources toward mnemonic processes[108-113]. During memory tasks, the functional coupling between HPC and VAN is enhanced[114,115], which in turn relates to mnemonic quality[116]. Moreover, the DA-A, VAN-A, and DMN-D include a large portion of the temporo-parietal junction (TPJ), which has been shown to be involved in contextual updating[117] and representing mental states[82]. Recent research suggests that the DMN and DA networks juxtapose near the TPJ—and that these networks serve as a hub for EC and HPC communication with other large-scale networks[80]. While speculative, it is possible that changes in the resting connectivity patterns of these networks may reflect changes in attentional shifts toward (or updating of) MTL contextual representations for memory.

### Beyond cortical–MTL interactions: global temporal drifts

Of note, while our results indicated a reasonable degree of regional specificity in the strength of the association between the passage of time and changes in EC- and aHPC-whole-brain resting connectivity patterns, we also found weak (but statistically significant) time-correlated drifts in our control ROIs (M1 and PRC). Current theoretical frameworks suggest that experience-driven and spontaneous synaptic plasticity may contribute to some degree of temporal drift over time[96,118,119]. Indeed, post-task resting state functional connectivity has been shown to strengthen in EC and HPC with memory consolidation[120-122] as well as in M1 after motor learning[123,124], suggesting that experience-dependent consolidation may drive system reconfiguration at rest. Relatedly, spontaneous and learning-associated changes in synaptic strength, spine density, and spine formation have been observed in both the EC and HPC[125-128] as well as in our control ROIs (PRC and M1)[129-133]. Given its known link to BOLD signal changes[134], synaptic plasticity may serve as the neurobiological substrate underlying the whole-brain functional connectivity changes observed in our study. While these studies underscore the ubiquity of dynamic changes throughout the brain, it is important to note that time-related changes in whole-brain connectivity patterns were significantly stronger in the EC and aHPC compared with our control regions (PRC and M1), whose functional connectivity drifts were instead better explained by other, time-correlated factors (here, hormonal changes) rather than the passage of time per se. Of note, neural excitability changes over time may also partially underlie the time-related changes in HPC- and EC-whole-brain resting connectivity that we observed in this study. Functional imaging[107] and cellular recording[17,27,105,106] studies examining long-term neural activity changes have revealed temporal drifts in local multivariate activation patterns over the course of weeks to months, even in the absence of explicit

task demands[16,19]. Determining whether regional differences in neuronal excitability and activity changes over time may contribute to stronger time-correlated drifts observed in EC and HPC resting connectivity patterns (compared to control regions) requires additional future investigation.

### Limitations and future directions

Our study used a dense-sampling approach whereby we examined $n = 2$ subjects (one Male and one Female) over a 30-day period to test whether time-related drifts occur in HPC- and EC-whole-brain functional connectivity patterns in the absence of task demands. Dense-sampling approaches have been recently advocated as a path forward to circumvent fMRI reliability issues that plague studies with modest sample sizes[59,135–137]. By increasing the number of measurements within subjects, dense sampling increases signal-to-noise ratio and statistical power to detect small effects sizes[59,135–137]. Dense sampling approaches also increase spatial precision in neuroimaging[137], which is especially important for brain regions known to have large cross-subjects variability[138]. Nonetheless, a tradeoff of this approach, compared to population-level effects, is generalizability to new subjects. Therefore, while all findings reported here were characterized independently in both a male and a female subject, future studies with larger sample sizes will be important to establish the generalizability of our findings to more diverse samples. In addition, it remains unclear whether and how the intrinsic resting connectivity metrics we examined relate to changes in stimulus-evoked neural activity patterns previously shown to vary with time[91]. Therefore, future work simultaneously examining time-related neural drifts in resting and task-based metrics in the same subjects—ideally with measurements of temporal memory—will be required to fully uncover the functional implications of the results uncovered here.

In conclusion, our study reveals a spontaneous neural signature that reflects the passage of time in humans in the absence of task demands, which may serve to provide temporal stamps for episodic memory processes.

## Methods

### Participants

Two healthy, right-handed adults (1 Female, 23 years; 1 Male, 26 years) with normal or corrected-to-normal vision participated. Both participants provided consent, and the study was approved by the University of California, Santa Barbara Human Subjects Committee[59,61].

### Experimental procedure

As previously described[59,61,62], the female subject underwent one session per day for 30 consecutive days with each session starting at 11:00 am. The male subject also underwent repeated testing for 30 days with the first 10 days consisting of testing at 7:00 am each day, the second 10 days consisting of testing at 7:00 am and 8:00 pm each day, and the final third 10 days consisting of testing at 8:00 pm each day (Fig. 1A). Serum and salivary assessments of hormones (Female: estradiol, progesterone, luteinizing hormone (LH) and follicle stimulating hormone (FSH), Male: estradiol, testosterone, cortisol) and questionnaires (Perceived Stress Scale, State-Trait Anxiety Inventory for Adults, and Profile of Mood States) were sampled at the start of each session. Of note, hormone samples from the female subject were collected with blood. In the male subject, the participant underwent one blood draw per day on days 11-20, whereas both saliva and blood sample for the hormone test were collected on days 1-10 and 21-30.

### fMRI acquisition

MRI scans were performed with a Siemens 3T Prisma scanner. For each session, participants completed a resting state fMRI scan with their eyes open (Female: 10 minutes, Male: 15 minutes, repetition time [TR] = 720 ms; echo time [TE] = 37 ms; 2.0-mm isotropic voxels, multiband

factor = 8), a $T_1$-weighted structural scan (TR = 2500 ms, TE = 2.31 ms, $T_1$ = 934 ms, 0.8 mm thickness) and a $T_2$-weighted hippocampal scan that was acquired with an oblique coronal orientation positioned orthogonally to the main axis of the hippocampus (TR = 8100 ms, TE = 8100, 0.4 ×0.4 mm² in-plane resolution, 2 mm slice thickness). Detailed scan parameter information can be found in refs. 59,60.

### Brain area parcellation

**Medial temporal lobe ROIs.** We parcellated medial temporal lobe (MTL) ROIs into HPC, EC, and perirhinal cortex using the automatic segmentation of hippocampal subfields package (ASHS)[60]. Briefly, $T_1$ and $T_2$ weighted images were submitted to ASHS, which automatically parcellates MTL using the Princeton Young Adult 3 T ASHS Atlas template[139]. Segmentations were then manually corrected using ITK-SNAP[140] based on the Olsen-Amaral-Palombo segmentation protocol[60,141]. All subfields of the hippocampus (including CA1, CA2/3, dentate gyrus, and subiculum) were merged into one hippocampal segmentation and then divided into anterior and posterior hippocampus according to the presence of uncus[36]. The masks were then manually inspected (and when needed, corrected) by a trained neuroanatomist (Dr. Jingyi Wang) (Supplementary Fig. 5). The EC was segmented using a well-validated parcellation strategy that uses anatomical landmarks[53,142]. Briefly, the most anterior level of the EC was fully covered by the alEC. We began delineating the pmEC at the very medial/dorsal tip of the EC, 2 mm after the appearance of the hippocampal head. The border between the alEC and pmEC gradually shifted laterally, forming an oblique boundary relative to the medial wall. Finally, we labeled all EC voxels as pmEC, where the uncus was no longer present[53,142–144]. Next, manually corrected MTL segmentations were registered to anatomical space (maintaining functional resolution) using FLIRT.

**Primary motor cortex (M1).** The M1 ROI was obtained from the Oxford PFC Consensus Atlas (http://lennartverhagen.com/[145,146]), thresholded at 25% and registered to participants' native surface space using Freesurfer[147]. Then, vertex coordinates in the M1 mask were transformed into the anatomical (volumetric) space. ROI masks in volumetric space were constructed by projecting half the distance of the cortical thickness at each vertex, with functional voxels required to be filled at least 50%. Volumetric masks were then resampled to functional resolution (2 mm³)[148].

**Network masks.** Seventeen cortical network masks were obtained using the Yeo-17 MNI atlas[83]. We registered each mask from MNI space to participants' anatomical space using FNIRT (10 mm warp resolution) while maintaining functional resolution (2 mm³).

### fMRI data processing and analysis

**Preprocessing.** We performed motion correction, skull removal, and registration of each individual's functional data to their anatomical space using the FMRIB Software Library (FSL). Six motion parameters and the average signal obtained from FSL-derived cerebrospinal fluid and white matter masks (obtained using FAST) were entered as nuisance regressors using AFNI's 3dDeconvolve function[149]. Next, a bandpass filter ($0.01 Hz < f < 0.1 Hz$) was applied using AFNI's 3dBandpass function[138]. The preprocessed functional data for each session and subject were kept in subject-specific T1 space and in the original functional resolution.

**Regional-whole-brain temporal drift score.** To obtain EC- and HPC-whole-brain resting state connectivity patterns for each session, we first averaged the time series across all voxels within each ROI. Pearson's correlation values (indexing resting-state functional connectivity with each seed) were then obtained using AFNI for all gray-matter voxels (for details, see Supplementary Methods) (Fig. 1B), which were

Fisher-Z transformed. Next, we calculated Pearson's correlation coefficients (i.e., indexing similarity) for resting-state functional connectivity patterns between every pair of sessions (across all gray matter voxels), which were Fisher-Z-transformed. Between-session similarity values were considered outliers and excluded from subsequent analyses if they exceeded 3 standard deviations relative to the mean across all pairs of sessions for each seed region [fewer than 1% session-pairs were excluded for each seed region, range of session pairs included: Female: 432–435 sessions (Mean = 434.087, sd = 0.848); Male: 769–780 sessions (Mean = 778.783, sd = 2.522)]. Finally, to capture whether connectivity patterns reliably tracked elapsed time, a temporal drift score was calculated for each seed ROI (Fig. 1D). To do so, we correlated the similarity of connectivity patterns (Z-transformed correlation coefficients obtained for every session pair) with the Δ time interval between session pairs. Finally, to isolate whether relationships with elapsed time were specific to particular large-scale cortical networks, temporal drift scores were calculated using connectivity patterns obtained from established large-scale cortical networks[83]. To do so, we divided the cortex into 17 networks using the Yeo-17 atlas[83]. For each session, we correlated the averaged time series of each seed region (e.g., EC, HPC, M1, and PRC) with the time series of every voxel within each network mask. We then obtained between-session pattern similarity by correlating connectivity patterns within each network mask *across* sessions. Finally, we obtained temporal drift scores by correlating between-session pattern similarity with the time interval between session pairs for each seed-network pair.

**HPC longitudinal axis and EC subregional analysis.** Temporal drift scores for the analysis of the HPC longitudinal axis were calculated as described above, but using single-voxels in the HPC as seeds, instead of the average timeseries in the ROI. Similarly, voxel-wise temporal drift scores in the EC were calculated and then averaged within each EC subregion.

### Statistical analysis

**Temporal drift score significance tests.** We performed two-tailed Pearson tests for the correlations between resting functional connectivity pattern similarity and time interval for each session pair for each ROI.

In addition, we also used a nonparametric approach to examine the significance of temporal drifts in a given ROI by testing against a null (chance) distribution. For nonparametric tests against chance, we assessed whether the true correlation was higher than the null distribution (rather than lower or higher), therefore, p-values are reported as one-tailed. To generate the null distribution, we permuted the time interval labels ($n = 5000$ times) and correlated the permuted labels with resting connectivity pattern similarity, which yielded a null distribution of temporal drift scores for a given ROI. The true temporal drift score for each ROI was then compared to each ROI's null distribution to estimate the permutation test p-value.

**Control analysis: test of regional differences in temporal drift strength.** To test whether our main ROIs showed stronger temporal drift than control ROIs, we used the Cocor package[150] to test for differences in temporal drift scores (i.e., test of the difference of dependent correlation coefficients) obtained from EC and HPC regions relative to the control regions (M1 and PRC). We also tested for differences in temporal drift by comparing EC- and/or HPC-network drift scores (for each network of interest) with temporal drift scores obtained using the network that served as a control (combined somatomotor A & B). Given the a priori prediction that pattern similarity would decrease over time, and thus more negative temporal drift scores would indicate stronger drifts, p-values for the comparison between correlation coefficients from a priori EC and HPC ROIs relative to control ROIs are reported at one-tailed $p < 0.05$.

**FDR correction.** When assessing temporal drift scores across the nodes of the Yeo-17 network, we corrected for multiple comparisons by adjusting the p-value using the Benjamini-Hochberg procedure across all ROIs[151].

**Linear gradient test.** To test whether temporal drift scores systematically vary along the longitudinal (y) axis of the HPC, we computed Pearson's correlation coefficients between temporal drift scores and the y-axis coordinates across all hippocampal voxels. To test for significance, we used a nonparametric permutation test. Specifically, to build a null distribution of correlation values, we permuted the y-axis coordinates of hippocampal voxels ($n = 5000$ times) and correlated the permuted y-axis coordinates with temporal drift scores across voxels (thus eliminating the true relationship between y-position and temporal drift scores). The true Pearson's correlation coefficient between voxel-wise drift scores and y-axis coordinates was compared to this null distribution to estimate the p-value for a putative longitudinal gradient.

**Kruskal-Wallis test.** To test whether temporal drift scores showed significant variance across the large-scale networks, we first obtained temporal drift scores from EC and aHPC with each of the Yeo-17 networks and computed Kruskal-Wallis H values (across female and male subjects). Next, to test for significance, we used a nonparametric permutation test (null distribution of H values obtained from $n = 5000$ permutations of time interval labels for EC and aHPC). Then, true H values obtained from EC and aHPC network analyses were compared to their respective null distributions to estimate p-values.

**Control analyses: time-varying factors.** To control for the effect of time-varying factors in our dataset—including hormonal, head motion, and emotion-related—we performed a multiple linear regression simultaneously entering changes in time interval, hormone, head motion, and emotion differences to predict changes in resting connectivity similarity between pairs of sessions. Hormonal changes (Female: estradiol, progesterone, LH and FSH, Male: estradiol, testosterone, cortisol) were calculated with the hormonal level sampled prior to each scan (Fig. 1A). Head motion changes were calculated as the difference of mean framewise displacement between pairs of sessions. We performed a principal component analysis (PCA, principal function, Psych R package[152]) using the three mood questionnaires (Perceived Stress Scale (PSS), State-Trait Anxiety Inventory for Adults (STAI), and Profile of Mood States (POM) tension and depression subscales) administered at the beginning of each session to obtain an aggregate metric of participant's emotional states for each session (First PC: Female subject: $\lambda_{pss} = 80.0\%$, $\lambda_{STAI} = 95.1\%$, $\lambda_{POM\_tension} = 94.2\%$, $\lambda_{POM\_depression} = 88.0\%$; Male subject: $\lambda_{pss} = 57.5\%$, $\lambda_{STAI} = 85.7\%$, $\lambda_{POM\_tension} = 48.4\%$, $\lambda_{POM\_depression} = 55.5\%$). Session-wise emotion differences were calculated as the difference between emotional-state scores for each pair of sessions.

### Reporting summary

Further information on research design is available in the Nature Portfolio Reporting Summary linked to this article.

## Data availability

The female subject dataset is openly available at https://openneuro.org/datasets/ds002674. The male subject dataset is openly available at https://openneuro.org/datasets/ds005115. The averaged functional connectivity maps for each ROI are shared in NeuroVault (https://neurovault.org/collections/VEHAFBWA/, Supplementary Figs. 6, 7). Source data are provided with this paper.

## Code availability

Analyses were run using custom code in FSL (version 6.0.7.12), Python (version 3.8, Package: Nilearn_0.9.1), AFNI (Version 24.1.22), and R (R

studio Version 1.4.1717, R version 4.1.2. Packages: tidyr_1.2.1; dplyr_1.0.10; emmeans_1.6.3; stats_4.1.2; cocor_1.1.3; ggplot2_3.4.2). The code used for these analyses is available in the following GitHub repository (https://github.com/LEAPNeuroLab/rsFCTemporalDrift)[153].

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

## Acknowledgements

This work was supported by the National Institute of Mental Health Grant R01-MH134000 (R.C.L.), a Hellman Fellows award (R.C.L.), the Ann S. Bowers Women's Brain Health Initiative (E.G.J., C.T.), NIH AG063843 (E.G.J.) and NIH AG079790 (L.P.). The authors thank Reicher Bergstein, Sydney Fortner, and Christina Villanueva for assistance with manuscript preparation. Icons in Fig. 1A, B were downloaded from the Noun Project (https://thenounproject.com/) and created by Ilham M. Rifai and Alzam. The published icons are licensed under the Creative Commons Attribution License (CC BY 3.0).

## Author contributions

R.C.L. and J.W. conceptualized and designed the experiment. J.W., A.T. and R.C.L. contributed to data analyses. L.P., C.M.T. and E.G.J. collected the data. All authors contributed to the writing of the final version of the manuscript.

## Competing interests

The authors declare no competing interests.
