## [Transparent Peer Review file · Nature Communications]

The intrinsic time tracker: Temporal context is embedded in entorhinal and hippocampal functional connectivity patterns

Corresponding Author: Dr Regina Lapate

Version 0:

Reviewer comments:

Reviewer #1

(Remarks to the Author)

The authors investigated whether changes in entorhinal cortex and hippocampal voxel-wise connectivity during rest drifted over a period of 30 days. Previous efforts have examined how multivariate patterns for stimulus-evoked activation drift over time. This study differs in that they looked at changes in seed-voxel connectivity patterns over time, in the absence of a task. The authors found evidence of drift in connectivity patterns in the entorhinal cortex and anterior hippocampus over time (i.e. the greater the distance between days, the less similar seed-voxel connectivity patterns were). This relationship between delay and pattern changes was stronger in the anterior hippocampus than the posterior hippocampus. The authors also detailed with which cortical networks the drift was driven by.

Overall, this is a compelling and well-executed study that will likely appeal to the broad readership of Nature Communications. The investigation of neural pattern drift is still in its early stages, and the authors' use of a dense sampling approach positions this work as an important contribution to the field. A key strength of the study is the robustness of its findings, even after controlling for potential confounding variables such as hormone and mood fluctuations over these extended timescales. The analyses were relatively straightforward, and the manuscript is well-written and clear.

My primary comments pertain to the interpretation of the data, which I believe could be refined and expanded upon:

1. The conclusion that temporal drift is specific to the EC and HC should be stated more cautiously. While the correlation between time delay and voxel-wise connectivity pattern similarity is stronger in these regions compared to control regions, the presence of drift in control regions—albeit weaker—raises questions. How should we interpret the evidence of drift in control regions? Could these regions also be “tracking” time, but on a different scale or to a lesser degree?
2. The authors' conclusion that EC and HC drift reflects a “tracking” of time may be overstated. The term “tracking” implies an active process, yet many uncontrolled factors could contribute to the observed drift (e.g., familiarity with the scanning environment, novelty effects, boredom, alertness). While the study shows that connectivity patterns change over time, it is not definitive evidence that these regions are actively tracking time.
3. The manuscript would benefit from further discussion about what the connectivity drift observed in this study conceptually represents. Studies of drift in humans thus far focus on local multivariate activation patterns. My understanding is that neuronal excitability changes over time might explain the phenomenon. Should we interpret changes in seed-voxel connectivity in the same way? Does the observed connectivity drift reflect local excitability changes in the cortex, or EC/HC or both?
4. How do the findings align with previous work cited in the manuscript that characterizes the anterior HC as having a coarser coding scheme, slower temporal autocorrelation, and gist-like memory representations? Intuitively, these properties might suggest less drift over time in anterior HC. Conversely, given the posterior HC's stronger connectivity to posterior cortical systems involved in context representation, wouldn't it be expected to exhibit greater drift in those connections over time? Clarifying this apparent discrepancy would strengthen the discussion.

Minor

- The description of the network analysis method could be clearer. Did the authors mask the seed-voxel connectivity maps

with the network of interest and then re-correlate them across sessions? This should be clarified in the methods section.
- What statistical package was used for the analyses? Including this detail would improve transparency.

Reviewer #2

(Remarks to the Author)

Wang et al. wrote an interesting paper describing novel results addressing the question of spontaneous time tracking in the human ERC and HIPP. The authors use HIPP and ERC masks to examine similarities in whole-brain and network correlations as a function of time difference between fMRI scanning sessions. The authors find that at the level of whole-brain correlations, greater temporal distance between sessions was associated with reduced similarities in whole-brain correlations with HIPP and ERC. Moreover, these correlations were localized to specific cortical networks, such as subdivisions of the DMN and DA. Finally, the authors point to a functional gradient along HIPP long-axis indicating differential involvement of anterior vs posterior HIPP in time tracking.

I really enjoyed reading the manuscript and I think it can be a nice contribution to the literature. The question is of high interest and relevance to the field, and the individualized approach to the study of time tracking is novel and original.

I do have several comments, mostly pertaining to the analyses and conceptualization of the ideas presented in the manuscript:

1. I really liked the individualized approach to the study of MTL function in general and to time tracking in particular. This study is a very nice example to the growing interest in the field of human neuroimaging in within-subject analyses. Having said that, it is unclear why the authors perform individualized neuroimaging, but use group-level atlases for defining MTL regions and cortical networks. The greatest advantage of individualized neuroimaging is to be able to take into account idiosyncratic anatomy to describe functional properties of interest in greater anatomical detail compared with using the currently dominant group-average approaches. Recent work from Thomas Yeo group has published multiple open-source algorithms for performing subject-specific multi-session cortical parcellations (Kong et al., 2019 *Cerebral Cortex*; Kong et al., 2022 *Cerebral Cortex*), which I believe will dramatically increase the anatomical specificity of the reported effects that are currently estimated using group-average atlases. Alternatively, the authors can use the (almost) original parcellation approach used in Yeo 2011 paper, but applied to individualized data (using the k-means clustering approach as presented in e.g., DiNicola et al., 2020 *Journal of neurophysiology*).

A similar issue arises with using an automatic segmentation procedure of ERC and PRC. MTL is highly variable across participants and using group-level templates might prevent from capturing subject-specific ERC and PRC anatomy. The authors mention on page 24 that the parcellations were manually corrected, but I could not find an image displaying the final mask used for ERC and PRC for each individual. Adding these final masks as a supplementary figure will be highly helpful.

2. The authors use the Yeo 17-Network solution for cortical organization. While this is indeed one of the most frequently used cortical parcellations in the field of human neuroimaging, the authors might want to consider a recently updated 15-network solution which was based on individualized, precision analyses (Du et al., 2024 *Journal of neurophysiology*). In any event, while using the 17-Network solution is totally valid, the network labels that the authors use are unclear to me. For example, to the best of my knowledge, Yeo 17-Network solution defines three DMN – DMN-A, DMN-B and DMN-C. It is unclear where does DMN-D mentioned by the authors come from. I believe the cortical network labeled as DMN-D in the manuscript is the temporal-parietal network in Yeo 17-Network solution, which was later recognized as the language network. Finally, while the authors use limbic networks A and B, I believe the recent paper by Girn et al., 2024, *Imaging Neuroscience* might be of interest. This study assigns regions of the limbic networks to DMN. I believe these recent data can help the authors to interpret some of their findings. Also please note that the updated 15-network solution (Du et al., 2024 *Journal of neurophysiology*) has no limbic networks.

3. The rationale for the study presented in the introduction is currently missing important prior work addressing the functional neuroanatomy of human MTL and its cortical interactions. (1) Interactions between ERC and HIPP are important since ERC is the main gateway of cortical input and output of HIPP (Witter et al., 1989, *Progress in Neurobiology*). (2) Cortical interactions between ERC and HIPP are important due to the network-level analysis performed by the authors; presenting this previous work (mentioned below) will make a stronger case for the performed analyses and connect the introduction better with the results and discussion. Previous studies associated subregions of the human ERC with different regions of CA1/subiculum and even more relevant to the current study – different regions of ERC with HIPP long-axis; furthermore, anterior and posterior HIPP were found to be associated with different cortical networks, similar to different regions of ERC that associate with different cortical networks/systems (Angeli et al., 2023 *bioRxiv*; Zheng et al., 2021 *PNAS*; Reznik et al., 2024 *Current Biology*; Reznik et al., 2023 *Neuron*; Maass et al., *eLife* 2015; Navarro Schröder et al., 2015 *eLife*). These and other studies (see below) should be mentioned as the authors build the rationale for the interactions between ERC and HIPP (including anterior vs posterior HIPP) and the interactions between ERC-HIPP and distributed cortical regions.

4. While the authors acknowledge the division of the HIPP into anterior and posterior HIPP, the ERC is analyzed and presented as one region without intrinsic divisions. While I understand the challenge in performing detailed MR imaging of human ERC, the authors should mention that ERC divisions in the rat, lateral ERC and medial ERC, are implicated in time coding (Eichenbaum, 2017 *Neuron*; Tsao et al., 2018 *Nature*). Recent investigations into the human ERC suggest that different divisions of human ERC are associated with different cortical networks pointing to different functional roles in

different subregions of human ERC. Similar to my previous comment, I believe addressing the functional and anatomical complexity of the ERC in humans and animals will build a stronger case for the rationale of the paper and the analyses performed by the authors.

5. To continue with my previous comment on a more fine-grain examination of ERC subregions, I suggest the authors to perform the voxel-wise analysis on the ERC as well (the same analysis the authors performed on the HIPP) and to examine the difference in temporal coding along the anterior-posterior ERC axis - reflecting potential human homologues of rodent lateral ERC and medial ERC (Maass et al., eLife 2015; Navarro Schröder et al., 2015 eLife) and along the medial-lateral ERC axis, reflecting the recently reported ERC functional bands (Reznik et al., 2024, Current Biology). I believe this is an important analysis to add to the manuscript in order to examine in more detail the functional properties of human ERC and to strengthen the authors' conclusions.

6. In page 20 the authors write that "Given that ERC is part of the DMN-C" and refer to Yet et al., 2011. To the best of my knowledge, Yeo et al do not report any associations between ERC and DMN. In fact, robust evidence for the association between ERC and different subdivisions of the canonical DMN (DMN-A and DMN-B) was provided only recently (Reznik et al., 2024, Current Biology; Reznik et al., 2023 Neuron).

7. Can the authors please plot the whole-brain correlations calculated using the ERC, HIPP, M1 and PRC seeds in a supplementary figure? I believe having these data displayed can be very helpful for examining "model-free" whole-brain associations of the MTL regions and to compare them to previous reports (Libby et al., 2012 Journal of Neuroscience; Kahn et al., 2008 Journal of neurophysiology and other studies mentioned earlier).

8. I think the authors can add another exciting analysis to the manuscript. While I leave this recommendation to the discretion of the authors, I believe this analysis can provide important insights into the time tracking mechanism. Since the male subject was scanned twice a day for 10 days, it opens the opportunity to examine the similarities between within-day vs between-days sessions to examine the role of sleep in spontaneous time tracking (Marshall and Born, 2007 Trends in cognitive sciences).

9. In page 27, the authors mention that they performed a control analysis to examine the differences in drift score between ERC/HIPP and M1/PRC, however these analyses are not directly referred to in the main text. I think these are critical comparisons and it is great that the authors performed these analyses, but I believe they should be mentioned in the main results section.

Version 1:

Reviewer comments:

Reviewer #1

(Remarks to the Author)

The authors have done an excellent job of addressing my concerns, and I have no further comments.

(Remarks on code availability)

I did not try to run the code but I did have a look. The authors provided code for the entire pipeline which is great. The code provides a readme file but it is a little sparse on instructions. For example, I don't see instructions on installing required software, directory structure required to run the code, variables that might need to be changed to get it to run, etc. The Readme file does contain information on the order the code should be run. The code itself is annotated which is good.

Reviewer #2

(Remarks to the Author)

I thank the authors for thoroughly addressing all my comments. I find it very interesting that aIEC showed stronger temporal drift than pmEC. This finding aligns well with stronger temporal drift in aHPC and overall, makes the manuscript more conceptually broad. I have only a few minor suggestions:

On page 4 the authors write that "The HPC can be divided into anterior and posterior hippocampus (corresponding to ventral and dorsal hippocampus in the rodent) which are preferentially connected to the medial and lateral subregions of the EC, respectively." And then the authors add "Extant data suggest that the strength of tracking of time in humans may be differentially supported by anterior versus posterior HPC as well as by lateral versus medial subregions of the EC." Medial and lateral subregions of the EC that are differently connected to the anterior vs posterior hippocampus are not the same "lateral versus medial subregions of the EC". Therefore, I suggest that the authors write "The HPC can be divided into anterior and posterior hippocampus (corresponding to ventral and dorsal hippocampus in the rodent) which are preferentially connected to different parts of the EC".

On page 15, the authors write "We found that the EC, aHPC, and aIEC ..." - aIEC is part of EC, so the phrasing is a bit odd.

Page 17, footnote 1 - many studies other than Braga/Reznik refer to DMN-C as DMN-A - e.g., DiNicola et al. (2020) Journal of neurophysiology; Angeli et al. (2025) PNAS; Du et al. (2024) Journal of neurophysiology. I believe that the authors can refer to the other naming convention (DMN-A) and provide the relevant references, similar to referring to the naming convention of the DMN-MTL system.

In the discussion (page 20), I believe it might be clearer to discuss first the EC role in the temporal drift and then mention EC divisions. Currently the authors mention aIEC and pmEC and then go back to EC. This might be confusing for the readers.

I have no further comments and I congratulate the authors on this great manuscript!

(Remarks on code availability)

We thank the Editor for the opportunity to revise our manuscript “*The intrinsic time tracker: Temporal context is embedded in entorhinal and hippocampal functional connectivity patterns*”, and the Reviewers for their thoughtful comments, questions, and suggestions. As a result of addressing them, we believe our revised manuscript has been greatly strengthened. The additional analysis following the Reviewers’ suggestions enriched our findings and helped us better contextualize them within a broader existing literature. Below, the Editor’s and the Reviewers’ comments are shown in indented *italicized* font, our replies are in regular 12pt. font, and changes to the manuscript are in regular 11pt. font and blue.

REVIEWER COMMENTS

Reviewer #1 (Remarks to the Author):

The authors investigated whether changes in entorhinal cortex and hippocampal voxel-wise connectivity during rest drifted over a period of 30 days. Previous efforts have examined how multivariate patterns for stimulus-evoked activation drift over time. This study differs in that they looked at changes in seed-voxel connectivity patterns over time, in the absence of a task. The authors found evidence of drift in connectivity patterns in the entorhinal cortex and anterior hippocampus over time (i.e. the greater the distance between days, the less similar seed-voxel connectivity patterns were). This relationship between delay and pattern changes was stronger in the anterior hippocampus than the posterior hippocampus. The authors also detailed with which cortical networks the drift was driven by.

Overall, this is a compelling and well-executed study that will likely appeal to the broad readership of Nature Communications. The investigation of neural pattern drift is still in its early stages, and the authors' use of a dense sampling approach positions this work as an important contribution to the field. A key strength of the study is the robustness of its findings, even after controlling for potential confounding variables such as hormone and mood fluctuations over these extended timescales. The analyses were relatively straightforward, and the manuscript is well-written and clear.

My primary comments pertain to the interpretation of the data, which I believe could be refined and expanded upon:

1. The conclusion that temporal drift is specific to the EC and HC should be stated more cautiously. While the correlation between time delay and voxel-wise connectivity pattern similarity is stronger in these regions compared to control regions, the presence of drift in control regions—albeit weaker—raises questions. How should we interpret the evidence of drift in control regions? Could these regions also be “tracking” time, but on a different scale or to a lesser degree?

We thank the reviewer for raising the important question of how to best interpret the presence of time-related drifts in our control ROIs—i.e., the perirhinal cortex (PRC) and primary motor cortex (M1)—and for giving us the opportunity to clarify our results. Below, we (a) provide evidence from additional control analyses that suggest that some of these broad changes, as seen in our control ROIs, may be primarily driven by ‘non-temporal’ factors; (b) discuss potential contributing mechanisms (e.g., learning and consolidation,

as well as synaptic plasticity) that we believe may underlie broad changes in resting-state connectivity patterns, including in our control ROIs.

First, we conducted additional analyses on our control regions to determine the extent to which time interval *per se* contribute to resting connectivity pattern changes between session pairs in the control ROIs after accounting for time-varying factors (i.e., changes in motion, emotional state, and hormone fluctuations) using a multiple regression framework. After accounting for these time-varying factors, we found that the similarity of whole-brain resting patterns no longer correlated with the time interval elapsed between sessions in *either* of our control regions (Female M1: $B = -0.0003$ (SE = 0.001), $t = 0.313$, $p = 0.754$; Perirhinal Cortex (PRC): $B = -0.004$ (SE = 0.001), $t = -0.446$, $p = 0.656$; Male M1: $B = 0.0002$ (SE = 0.002), $t = 0.09$, $p = 0.928$; PRC: $B = -0.001$ (SE = 0.002), $t = -0.411$, $p = 0.681$). Instead, sex hormone fluctuations remained the only significant predictor in this multiple regression model (**Supplementary Figure 3**), consistent with prior findings suggesting that hormonal fluctuations reliably predict broad changes in resting connectivity (Pritschet et al., 2020; Grotzinger et al., 2024). These results contrast with the results from the analogous control analysis conducted in the hippocampus (HPC) and entorhinal cortex (EC), wherein *time interval* continued to be a significant predictor of resting connectivity changes in EC and anterior HPC (in both subjects), even *after* accounting for time-varying changes in motion, emotion, and hormones (results previously reported in the main manuscript; Supplementary Figure 2).

Supplementary Figure 3. Multiple regression of time-varying factors in control ROIs. Only hormonal fluctuations (and not time interval) were associated with changes in similarity in M1- and PRC- whole-brain resting connectivity patterns in a simultaneous regression model that also included changes in emotion and head motion—in contrast to findings in EC and HPC (**Figure Supplementary Figure 2**). Fixed-effect estimates (β) are plotted with standard error (SE) bars for the female (left) and male (right) subjects. β and SE were standardized for visualization purposes. Abbreviations: LH: luteinizing hormone, FSH: follicle stimulating hormone. * Denotes statistical significance at $p \leq 0.05$; ** Denotes statistical significance at $p \leq 0.01$; *** Denotes statistical significance at $p \leq 0.001$.

Nonetheless, learning and consolidation may be linked to changes in whole-brain functional connectivity patterns observed in our control regions (Harmelech and Malach, 2013; Harmelech et al., 2013; Rule et al., 2019; Yu et al., 2020; Driscoll et al., 2022). For instance, a prior study has shown that working memory training can increase the network similarity in resting-state functional connectivity patterns between naive subjects and memory athletes, suggesting that learning induces measurable changes in brain network reconfiguration at rest using MRI-based metrics (Dresler et al., 2017). Post-task resting-state functional connectivity metrics have also been shown to be sensitive to memory consolidation processes (Tambini et al., 2010; Tomparry et al., 2015; Tambini and D’Esposito, 2020) and motor learning (Albert et al., 2009; Sami and Miall, 2013; Li et al., 2018). Thus, it is possible that learning and consolidation-related changes in whole-brain functional connectivity patterns occur throughout the study. These changes would result in greater similarity between whole-brain functional connectivity patterns associated with subtle learning-related changes (i.e., those that are more proximal in time) compared to those encompassing multiple learning-related events (i.e., those that are more distant in time). Thus, a certain degree of time-correlated drift may be expected in control ROIs (and much of the brain), given that experience-dependent plasticity (i.e., learning and consolidation) likely occurs throughout the brain over time.

Synaptic plasticity, a key mechanism of learning and memory (Harris, 1999; Harmelech and Malach, 2013; Harmelech et al., 2013; Segal, 2017; Moyer and Zuo, 2018; Magee and Grienberger, 2020), may underlie resting-state drifts throughout much of the brain, including in regions that served as control ROIs. For instance, following reward-association learning, prior studies have shown changes in synaptic strength and reduction

of spine density in PRC (Massey et al., 2008; Banks et al., 2012). Similarly, motor learning induces persistent changes in dendritic spine formation in M1 (Xu et al., 2009; Hayashi-Takagi et al., 2015). Beyond experience-driven synaptic plasticity, *spontaneous* spine formation in the M1 has also been found in adult mice (Yang et al., 2009). Notably, simultaneous extracellular recordings and fMRI scans in the rat HPC showed that inducing LTPs with high-frequency stimulation in the perforant path enhanced BOLD signals alongside increased neurophysiological indices for synaptic plasticity—such as excitatory postsynaptic potentials (Canals et al., 2009). This effect persisted for hours, suggesting a close relationship between synaptic plasticity and BOLD signal changes at rest. Thus, continuous circuitry rewiring inherent to synaptic plasticity likely contributes to the measurable large-scale drifts found in resting-state connectivity patterns.

Taken together, our control analyses and the studies reviewed above, suggest that while some degree of drift occurs over time in the PRC and M1, factors such as spontaneous synaptic plasticity, experience-dependent plasticity, and hormonal processes may better explain the variance in resting connectivity pattern changes (rather than the passage of time itself). Nonetheless, we agree with the Reviewer that their correlation with time interval warrants additional discussion in our manuscript as well as more caution in our conclusion regarding the specificity of time-related drifts to EC and HPC; after all, the factors described above (learning-driven consolidation, spontaneous and learning-drive synaptic changes, and hormonal fluctuations) are inherently—even if weakly—correlated with time.

Following the Reviewer's suggestion, we have (a) softened the language regarding the regional specificity of temporal drifts; (b) added a paragraph in the discussion about the potential sources of drifts, including in our control ROIs, and (c) revised our results (and Supplementary Materials) to include the above-mentioned additional control analysis in our control ROIs:

(A) Revised the language to state claims of regional specificity more cautiously:

Page 14 (Results):

Comparing the strength of time-related drifts in EC and HPC versus control regions

Next, we tested whether time-related drifts in whole-brain resting connectivity patterns are generally present when using other regions as seeds, *versus whether they are statistically stronger in EC- and aHPC- resting connectivity patterns compared to other regions. To that end,*

we compared the strength of their time correlation against two control regions: the primary motor cortex (M1) and the perirhinal cortex (PRC).

Page 16 (Results):

In sum, these results **suggest regional specificity in the strength of time-related EC and aHPC-whole-brain resting connectivity changes.**

Page 20 (Discussion):

Critically, time-related changes in whole-brain connectivity patterns were **significantly stronger in the EC and aHPC when compared with control regions (motor cortex and nearby medial temporal lobe (MTL) cortex)...**

Page 33 (Methods):

*Control Analysis: **test of regional differences in temporal drift strength.** To test whether our main ROIs showed stronger temporal drift than control ROIs, we used the Cocor package (Diedenhofen and Musch, 2015) to test for difference in temporal drift scores ...*

(B) Extended the discussion about the potential sources of drifting in control ROIs:

Page 26 (Discussion):

Beyond cortical–MTL interactions: Global temporal drifts

Of note, while our results indicated a reasonable degree of regional specificity in the strength of the association between the passage of time and changes in EC- and aHPC-whole brain resting connectivity patterns, we also found weak (but statistically significant) time-correlated drifts in our control ROIs (M1 and PRC). Theoretical works suggest that experience-driven and spontaneous synaptic plasticity may contribute to some degree of temporal drift over time (Harmelech and

Malach, 2013; Rule et al., 2019; Driscoll et al., 2022). Indeed, post-task resting-state functional connectivity has been shown to strengthen in EC and HPC with memory consolidation (Tambini et al., 2010; Tompary et al., 2015; Tambini and D'Esposito, 2020) as well as in M1 after motor learning (Albert et al., 2009; Sami and Miall, 2013), suggesting that experience-dependent consolidation may drive system reconfiguration at rest. Relatedly, spontaneous and learning-associated changes in synaptic strength, spine density, and spine formation have been observed in both the EC and HPC (Harris, 1999; Attardo et al., 2015; Segal, 2017; Moyer and Zuo, 2018) as well as in our control ROIs (PRC and M1) (Massey et al., 2008; Xu et al., 2009; Yang et al., 2009; Banks et al., 2012; Hayashi-Takagi et al., 2015). Given its known link to BOLD signal changes (Canals et al., 2009), synaptic plasticity may serve as the neurobiological substrate underlying the whole-brain functional connectivity changes observed in our study. While these studies underscore the ubiquity of dynamic changes throughout the brain, it is important to note that time-related changes in whole-brain connectivity patterns were significantly stronger in the EC and aHPC when compared with our control regions (PRC and M1), whose functional connectivity drifts were instead better explained by other, time-correlated factors (here, hormonal changes) rather than the passage of time *per se*.

(C) revised our results to add the additional analysis for control ROIs:

Page 17:

Conversely, after controlling for these factors, the similarity of whole-brain resting patterns in either of our control regions no longer correlated with the time interval elapsed between sessions (Female: M1: $B = -0.0003$ (SE = 0.001), $t = 0.313$, $p = 0.754$; PRC: $B = -0.004$ (SE = 0.001), $t = -0.446$, $p = 0.656$; Male: M1: $B = 0.0002$ (SE = 0.002), $t = 0.09$, $p = 0.928$; PRC: $B = -0.001$ (SE = 0.002), $t = -0.411$, $p = 0.681$). Instead, hormonal fluctuations remained the only significant predictor in this multiple regression model (**Supplementary Figure 3**), consistent with prior findings showing their robust influence on resting state connectivity patterns (Pritschet et al.,

2020; Grotzinger et al., 2024). Collectively, these results indicate that changes in EC- and aHPC-whole-brain resting functional connectivity patterns correlate with the passage of time beyond changes in neural similarity driven by other time-varying factors.

2. The authors' conclusion that EC and HC drift reflects a "tracking" of time may be overstated. The term "tracking" implies an active process, yet many uncontrolled factors could contribute to the observed drift (e.g., familiarity with the scanning environment, novelty effects, boredom, alertness). While the study shows that connectivity patterns change over time, it is not definitive evidence that these regions are actively tracking time.

We appreciate the Reviewer's invitation to revisit our terminology. While global connectivity pattern changes observed during resting-state scans may suggest the presence of a neural time-tracking mechanism, we acknowledge that this alone cannot provide definitive evidence that the time-related neural pattern changes observed in our study reflect an active process. As noted in our discussion (Page 28, below), we previously acknowledged this limitation and emphasized the importance of incorporating both resting-state and task-based scans in future studies.

Page 28 (Discussion):

In addition, it remains unclear whether and how the intrinsic resting connectivity metrics we examined relate to changes in stimulus-evoked neural activity patterns previously shown to track with time (Buonomano et al., 2023). Therefore, future work simultaneously examining time-related neural drifts in resting and task-based metrics in the same subjects—ideally with measurements of temporal memory—will be required to fully uncover the functional implications of the results uncovered here.

Moreover, we would like to clarify that our analytical strategy includes a number of time intervals sampled homogeneously from the 30-day period: for instance, time interval=1 or time interval=2 is sampled equally (and are equally represented) throughout our experiment. Thus, although familiarity (and hence novelty, and boredom) with the scanner environment naturally changes from the start to the end of the study, our study design mitigates the potentially confounding influence of time-related experiences. For instance, although alertness is likely to vary and correlate across consecutive days over time,

presumably alertness should not systematically vary *with* time (i.e., be necessarily maximally dissimilar between days 1 and 30). Nonetheless, following the Reviewer’s concern, we conducted an additional control analysis including “session order” (i.e., session 1, session 2, etc.) as a predictor. We found that time interval remained a significant predictor after controlling for session order in our main ROIs (see **Figure R1**), suggesting that changes in EC and aHPC whole-brain functional connectivity patterns reflect time intervals, even when accounting for session order that might more closely covary with (ordinal) session-varying factors such as familiarity, novelty, and boredom.

Importantly, in alignment with the Reviewer’s concern, we did not intend to imply that our data spoke to an ‘active’ tracking process—rather that they are reliably correlated with time (above and beyond time-varying factors), which is consistent with the possibility that they may track (as in *reflect*) the passage of time. We therefore have revised our manuscript to clarify what we meant by “tracking” and soften the statements that used that phrasing to emphasize that we are referring to the reliable correlation between the EC and HPC resting functional connectivity patterns and time interval (above and beyond time-varying potentially confounding factors measured in this study).

Figure R1. Multiple regression of time-varying factors. The passage of time is associated with reduced similarity in EC- and aHPC- whole-brain resting connectivity patterns after controlling for

changes in head motion, emotion, time interval, and session order. Fixed-effect estimates (β) are plotted with standard error (SE) bars for the female (left) and male (right) subjects. β and SE are standardized for visualization purposes. Abbreviations: LH: luteinizing hormone, FSH: follicle stimulating hormone. * Denotes statistical significance at $p \leq 0.05$; ** Denotes statistical significance at $p \leq 0.01$; *** Denotes statistical significance at $p \leq 0.001$.

Page 7:

To foreshadow, we found that changes in intrinsic EC and HPC-whole brain resting connectivity patterns **reflected** the passage of time in a task-free context in humans...

Page 16:

Collectively, these results **indicate** that changes in EC- and aHPC-whole-brain resting functional connectivity patterns **correlate with** the passage of time beyond changes in neural similarity driven by other time-varying factors.

Page 20:

Collectively, these data suggest that EC- and aHPC-whole-brain resting functional connectivity patterns spontaneously **reflect** the passage of time in humans.

Page 20:

EC and aHPC whole-brain connectivity patterns **reflect the passage of time**

Page 22:

Collectively, these findings suggest that changes in the similarity of EC and aHPC-whole brain functional connectivity patterns spontaneously **reflect** the passage of time.

Page 23:

Moreover, our finding that the strength of time-related drift in HPC signals covaried linearly with the position along the HPC longitudinal axis provides novel evidence for a potential anterior-posterior functional gradient of time coding along the HPC.

Finally, following our in-text clarification of what we meant by “tracking” (i.e., tracking with or reflecting), we have opted not to change the manuscript title (*The intrinsic time tracker: Temporal context is embedded in entorhinal and hippocampal functional connectivity patterns*). However, we would be willing to revise it if the Reviewer prefers.

3. The manuscript would benefit from further discussion about what the connectivity drift observed in this study conceptually represents. Studies of drift in humans thus far focus on local multivariate activation patterns. My understanding is that neuronal excitability changes over time might explain the phenomenon. Should we interpret changes in seed-voxel connectivity in the same way? Does the observed connectivity drift reflect local excitability changes in the cortex, or EC/HC or both?

In alignment with the Reviewer’s suggestion, we believe that neural excitability changes may potentially underlie the changes in seed-voxel connectivity that we observed in our study as suggested in recent theoretical proposals and studies investigating temporal drifts in neural activity patterns (Driscoll et al., 2022; Zvi and Elisha, 2023). Functional imaging studies in humans (Zvi and Elisha, 2023) and neuronal recordings in mice (Ziv et al., 2013; Mau et al., 2018; Marks and Goard, 2021; Schoonover et al., 2021; Xia et al., 2021) show that neuronal representations of the same stimuli change across experimental sessions, a phenomenon driven by changes in neuronal firing rates and neural excitability. Notably, place-cell activity pattern drifts in the absence of task demands (i.e. during an open-field task) suggests that temporal drifts may occur even without explicit task demands (Mankin et al., 2012, 2015). In the context of our experiment (i.e., task-free resting-state scans), episodic information, such as the temporal context of each experimental session, may be partially represented by local multivariate activation patterns that are similarly subject to drifts over time. This drift may be particularly prominent in EC and HPC due to their established roles in episodic context representations (Clewett et al., 2019; Sugar and Moser, 2019), further suggesting that time-related neural excitability changes may vary across brain regions.

Moreover, as discussed in response to this Reviewer's Comment #1, factors such as spontaneous synaptic plasticity, experience-dependent plasticity, and hormonal processes may also produce some degree of time-related neural pattern changes across the brain [including in our a-priori MTL and control ROIs (Harris, 1999; Albert et al., 2009; Tambini et al., 2010; Attardo et al., 2015; Tompary et al., 2015; Segal, 2017; Moyer and Zuo, 2018; Chen et al., 2020)]. While neural changes occur globally, their rate may vary across brain regions depending on the stimulus source [e.g., after an object-face encoding task, increased resting-state functional connectivity strength is specific to HPC-lateral occipital complex (Tambini et al., 2010)] and underlying mechanisms [e.g., spontaneous synaptic plasticity is stronger in the HPC than the cortex (Attardo et al., 2015)]. It is possible these regional differences in neuronal excitability and activity changes over time may, in turn, contribute to the time-correlated drifts in EC and HPC functional connectivity patterns observed in our study. That said, while theoretical models of neural temporal drift exist (Rule et al., 2019; Driscoll et al., 2022), its underlying mechanisms remain underexplored. Thus, future research is needed to fully elucidate the factors driving this phenomenon.

Following the Reviewer's suggestion, we have added further discussion about the putative mechanisms underlying resting functional connectivity changes over time, and to highlight that future studies will be needed to fully unveil them:

Page 28:

Of note, it is possible neural excitability changes over time may also partially underlie the time-related changes in HPC and EC whole-brain resting connectivity that we observed in our study. Functional imaging (Zvi and Elisha, 2023) and cellular recording (Ziv et al., 2013; Mau et al., 2018; Marks and Goard, 2021; Xia et al., 2021) studies examining long-term neural activity changes have revealed temporal drifts in local multivariate activation patterns over the course of weeks to months, even in the absence of explicit task demands (Mankin et al., 2012, 2015). Determining whether regional differences in neuronal excitability and activity changes over time may contribute to stronger time-correlated drifts observed in EC and HPC functional connectivity patterns (compared to control regions) requires additional future investigation.

4. How do the findings align with previous work cited in the manuscript that characterizes the anterior HC as having a coarser coding scheme, slower temporal autocorrelation, and gist-like memory representations? Intuitively, these properties might suggest less drift over time in anterior HC. Conversely, given the posterior HC's stronger connectivity to posterior cortical systems involved in context representation, wouldn't it be expected to exhibit greater drift in those connections over time? Clarifying this apparent discrepancy would strengthen the discussion.

We thank the Reviewer for the invitation to contextualize our findings within the broader literature of coding properties along the longitudinal axis of the HPC. We believe that our findings of stronger anterior hippocampal temporal drifts are consistent with known functional properties of the anterior (vs. posterior) HPC [i.e., coarse coding, slower temporal autocorrelation, and gist-like memory representation (Keinath et al., 2014; Strange et al., 2014; Collin et al., 2015; Brunec et al., 2018; Raut et al., 2020; Bouffard et al., 2023)], as follows:

First, our “temporal drift score” was calculated using *days* as a unit of time, a relatively coarse temporal metric relative to extant work. Prior work has shown that the posterior HPC represents fine-grained temporal information on the scale of minutes or seconds (Umbach et al., 2020; Reddy et al., 2021; Shimbo et al., 2021), a temporal resolution that is better captured by examining narrower time windows than the current experiment permits. Conversely, anterior hippocampal function has been proposed to have a relatively coarser coding scheme, including tracking conjunctive spatial-temporal representations in the units of days [e.g. over a 30-day period, (Nielson et al., 2015)].

Second, regions with faster temporal autocorrelation, such as the posterior HPC, may show less prominent temporal drift over longer experimental timescales (e.g., here, 30 days). For instance, when examined over a longer (18-month) time scale, it appears that the similarity of the whole-brain resting state connectome may ‘resets’ periodically, rather than becoming (ever) progressively dissimilar [e.g. citation Figure 3A of (Poldrack et al., 2015)]. This suggests that the extent to which temporal drift alone scales with time may be ‘time-limited’. A faster temporal autocorrelation rate in the posterior HPC may contribute to more frequent drift cycles, causing neural activity patterns to reset to their previous states over shorter temporal intervals. Consistent with this idea, prior studies have found that multivariate neural representations in the posterior HPC are less sensitive to detecting temporal changes over extended timescales (Nielson et al., 2015; Dandolo and Schwabe, 2018; Krenz et al., 2023).

We had previously discussed our findings on the functional gradient along the longitudinal axis of the HPC on Page 23, which we now further extend by incorporating the above

discussion to strengthen our interpretation of the stronger temporal drift observed in the anterior HPC:

Page 23:

Moreover, the relatively more rapid decline in the rate of temporal autocorrelation previously observed in the pHPC suggests that its neural activity patterns may revert to previous states more rapidly, potentially limiting systematic time-correlated drift over extended time periods (Brunec et al., 2018; Bouffard et al., 2023).

Minor

- The description of the network analysis method could be clearer. Did the authors mask the seed-voxel connectivity maps with the network of interest and then re-correlate them across sessions? This should be clarified in the methods section.

Following the Reviewer's suggestion, we have elaborated on our description of the network analysis methods.

Page 33:

To do so, we divided the cortex into 17 networks using the Yeo-17 atlas (Yeo et al., 2011). For each session, we correlated the averaged time series of each seed region (e.g., EC, HPC, M1, and PRC) with the time series of every voxel within each network mask. We then obtained between-session pattern similarity by correlating connectivity patterns within each network mask across sessions. Finally, we obtained temporal drift scores by correlating between-session pattern similarity with the time interval between session pairs for each seed-network pair.

- What statistical package was used for the analyses? Including this detail would improve transparency.

We clarified this in our "Data, Materials, and Code availability" as follows:

Page 37:

Data, Materials, and Code Availability

The female subject dataset is openly available at <https://openneuro.org/datasets/ds002674>. The male subject dataset is openly available at <https://openneuro.org/datasets/ds005115>. Analyses were run using custom code in FSL (version 6.0.7.12), Python (version 3.8, Package: Nilearn_0.9.1), and R (R studio Version 1.4.1717, R version 4.1.2. Packages: tidyr_1.2.1; dplyr_1.0.10; emmeans_1.6.3; stats_4.1.2; cocor_1.1.3; ggplot2_3.4.2), which is available in GitHub repository (<https://github.com/LEAPNeuroLab/rsFCTemporalDrift>). The averaged functional connectivity maps for each ROI are shared in NeuroVault (<https://neurovault.org/collections/VEHAFBWA/>, Supplementary Figure 6-7).

Reviewer #2 (Remarks to the Author):

Wang et al. wrote an interesting paper describing novel results addressing the question of spontaneous time tracking in the human ERC and HIPP. The authors use HIPP and ERC masks to examine similarities in whole-brain and network correlations as a function of time difference between fMRI scanning sessions. The authors find that at the level of whole-brain correlations, greater temporal distance between sessions was associated with reduced similarities in whole-brain correlations with HIPP and ERC. Moreover, these correlations were localized to specific cortical networks, such as subdivisions of the DMN and DA. Finally, the authors point to a functional gradient along HIPP long-axis indicating differential involvement of anterior vs posterior HIPP in time tracking.

I really enjoyed reading the manuscript and I think it can be a nice contribution to the literature. The question is of high interest and relevance to the field, and the individualized approach to the study of time tracking is novel and original.

I do have several comments, mostly pertaining to the analyses and conceptualization of the ideas presented in the manuscript:

IA. I really liked the individualized approach to the study of MTL function in general and to time tracking in particular. This study is a very nice example to the growing interest in the field of human neuroimaging in within-subject analyses. Having said that, it is unclear why the authors perform individualized neuroimaging, but use group-level atlases for defining MTL regions and cortical networks. The greatest advantage of individualized neuroimaging is to be able to take into account idiosyncratic anatomy to describe functional properties of interest in greater anatomical detail compared with using the currently dominant group-average approaches. Recent work from Thomas Yeo group has published multiple open-source algorithms for performing subject-specific multi-session cortical parcellations (Kong et al., 2019 Cerebral Cortex; Kong et al., 2022 Cerebral Cortex), which I believe will dramatically increase the anatomical specificity of the reported effects that are currently estimated using group-average atlases. Alternatively, the authors can use the (almost) original parcellation approach used in Yeo 2011 paper, but applied to individualized data (using the k-means clustering approach as presented in e.g., DiNicola et al., 2020 Journal of neurophysiology).

We appreciate the Reviewer's positive feedback on our dense sampling approach and for bringing to our attention these recently-developed individualized network parcellation methods. First, we would like to clarify that individual anatomy (i.e., each subject's native T1/T2 space) was used to define all MTL masks, which were manually inspected (and

modified as needed) by the first author (Dr. Wang), a trained neuroanatomist (see response to this Reviewer's Comment 1B for further details).

Second, we would like to clarify that the network analysis in our manuscript were primarily a supplemental (secondary) analysis mainly meant to identify and localize the drivers of the primary findings of our manuscript—i.e., EC and HPC time-related changes in whole-brain functional connectivity patterns.

Nonetheless, we implemented an individualized network parcellation following the Reviewer's suggestion (Kong et al., 2019, 2021) to calculate network-level temporal drift scores for the EC and aHPC in each participant (**Supplementary Figure 4**; included below for your convenience).

Our individualized network results largely aligned with our previous group-level network parcellation findings for EC-network connectivity changes (**Supplementary Figure 4**). Specifically, using the individualized-network masks, we found that EC temporal drifts were primarily driven by the DMN-C (Female: $r = -0.209$, $p_{FDR} < 0.001$; Male: $r = -0.229$, $p_{FDR} < 0.001$), DMN-D (Female: $r = -0.163$, $p_{FDR} = 0.002$; Male: $r = -0.196$, $p_{FDR} < 0.001$), dorsal attention network A (DA-A) (Female: $r = -0.179$, $p_{FDR} = 0.001$; Male: $r = -0.2$, $p_{FDR} < 0.001$), dorsal attention network B (DA-B) (Female: $r = -0.174$, $p_{FDR} = 0.001$; Male: $r = -0.174$, $p_{FDR} < 0.001$), ventral attention network-A (VAN-A) (Female: $r = -0.156$, $p_{FDR} < 0.001$; Male: $r = -0.170$, $p_{FDR} < 0.001$), and Visual-B (Female: $r = -0.177$, $p_{FDR} < 0.001$; Male: $r = -0.278$, $p_{FDR} < 0.001$). These findings largely align with our original results, which had implicated networks DMN-C, DMN-D, and DA-A. Notably, these networks showed significantly stronger temporal drift than the somatomotor network (i.e., our control network) in both subjects (EC-networks vs. control $ps < 0.05$), with the exception of the EC-DMN-D for the Female subject ($p = 0.088$).

However, when inspecting the individualized network-level results for aHPC, we noticed discrepancies between the two parcellation methods. We found that aHPC temporal drifts were driven by the VAN-A (Female: $r = -0.302$, $p_{FDR} < 0.001$; Male: $r = -0.209$, $p_{FDR} < 0.001$) and the Visual network (Visual-B) (Female: $r = -0.314$, $p_{FDR} < 0.001$; Male: $r = -0.235$, $p_{FDR} < 0.001$), similar to our original results, in which aHPC's connectivity change was driven by VAN-A and Visual-A. Note, however, these results were only significantly stronger than the control network in the Male subject (aHPC-VAN-A vs. control $p < 0.001$; aHPC-Visual-B vs. control $p < 0.001$), but not the Female subject ($ps > 0.083$). We also noticed some discrepancy in the individualized networks associated with aHPC relative to the group method—such as the emergence of the DA-B network for both subject (Female: $r = -0.274$, $p_{FDR} < 0.001$, aHPC-DA-B vs. control $p = 0.003$; Male: $r = -0.230$, $p_{FDR} < 0.001$, aHPC-DA-B vs. control $p = 0.027$) and the limbic network B (LIM-B) for the female subject only (Female: $r = -0.290$, $p_{FDR} < 0.001$, aHPC-LIM-B vs. control $p = 0.007$).

Lastly, we assessed the test-retest reliability of the individualized network parcellation in our two subjects using a split-half approach. As indicated by the arrows in Figure R2, we found that the network boundaries were not fully consistent across the two halves.

Given the supplemental nature of the original network analysis—and that this was the first study seeking to connect whole-brain hippocampal and entorhinal cortical resting connectivity drifts with the passage of time—we opted to keep our original Yeo-based network analysis in the main manuscript to maximize reproducibility for future investigators. However, we agree with the Reviewer regarding the importance of encouraging future investigators to further refine and embrace individualized approaches; therefore, we include the individualized network-level results shown above in the Supplementary Material—while also ensuring that our results remain accessible and replicable within a widely used group-level framework (i.e., the Yeo atlas).

Page 19:

Of note, these results were largely replicated when using an individualized network parcellation method (Kong et al., 2019, 2021) (for details, see Supplementary Results: Individualized network parcellation analysis).

Supplementary Figure 4. Individualized network analysis: EC and aHPC time-dependent pattern changes were driven by specific networks. Seventeen individually parcellated brain networks are shown on a surface template (color-coded by network) for each participant (Kong et al., 2019). The primary networks that drove time-related resting connectivity pattern changes in the **(A)** EC and **(B)** aHPC—using the individualized network parcellation approach—are highlighted. Bar plots show temporal drift scores for **(C)** EC and the individualized Yeo-17 large-scale networks (Female: left, Male: right) and **(D)** aHPC and the individualized Yeo-17 large-scale networks (Female: left, Male: right). *Denotes significantly different temporal drift scores compared to the control somatomotor network in each subject. ~Denotes temporal drift scores were at trend level different than the control somatomotor network in each subject. Abbreviations: default mode network (DMN), dorsal attention network (DA), ventral attention network (VAN), somatomotor network (SMotor).

Figure R2. Split-half (earlier vs. later sessions) individualized network mask parcellations for both subjects. Arrows indicate the intra-individual discrepancies across the two halves.

1B. A similar issue arises with using an automatic segmentation procedure of ERC and PRC. MTL is highly variable across participants and using group-level templates might prevent from capturing subject-specific ERC and PRC anatomy. The authors mention on page 24 that the parcellations were manually corrected, but I could not find an image displaying the final mask used for ERC and PRC for each individual. Adding these final masks as a supplementary figure will be highly helpful.

We appreciate the Reviewer's thoughtful comment and agree that the subregions of the medial temporal lobe (MTL) show substantial inter-individual variability, which was accounted for by several complementary strategies in the current parcellation method, as follows:

First, we employed the automatic segmentation of hippocampal subfields (ASHS) pipeline, a validated and well-established method (Palombo et al., 2013; Berron et al., 2017; Brunec et al., 2020; Taylor et al., 2020) that provides high accuracy relative to manual segmentation (Yushkevich et al., 2006, 2015a). This pipeline used the Princeton Young Adult 3T Atlas [N=24, mean age 22.5 years; (Aly and Turk-Browne, 2016)], which closely matched our participants' age range (Female: 23 years, Male, 26 years). Given the validity, reliability, and widespread usage of this pipeline (Palombo et al., 2013; Berron et al., 2017; Brunec et al., 2020; Taylor et al., 2020), we reasoned that adopting it would make our study more easily reproducible by others.

That said, the ASHS-generated masks were manually examined (and when needed, corrected) by a trained neuroanatomist, Dr. Jingyi Wang (the first author of this study) using the Olsen-Amaral-Palombo (OAP) protocol (Olsen et al., 2013; Palombo et al., 2013; Yushkevich et al., 2015b, 2015a). The OAP protocol was developed using a large number of high-resolution structural MRI and histological studies (Insausti et al., 1998; Pruessner et al., 2002; Insausti and Amaral, 2004; Frankó et al., 2014). Manual corrections were guided by anatomical landmarks (e.g., uncus, rhinal sulcus, collateral sulcus, and frontotemporal juncture), resulting in a few needed corrections (approximately 10% of originally labeled voxels changed) and ensuring precise MTL segmentation for each participant. We clarified that these masks were inspected and modified by the first author of the paper:

Page 30:

All subfields of the hippocampus (including CA1, CA2/3, dentate gyrus, and subiculum) were merged into one hippocampal segmentation and then divided into anterior and posterior

hippocampus according to the presence of uncus (Strange et al., 2014); masks were manually inspected (and when needed, corrected) by a trained neuroanatomist (Dr. Jingyi Wang) (Supplementary Figure 5).

We had previously shown the MTL mask used for one participant (female) in Figure 1C. Following the Reviewer's comment, we have now added a supplementary figure that displays the final MTL masks for both individuals:

Supplementary Figure 5. The medial temporal lobe masks are shown for the female (A) and male (B) participants. Abbreviations: aHPC: anterior hippocampus, pHPC: posterior hippocampus, aIEC: anterolateral entorhinal cortex, pmEC: posteromedial entorhinal cortex, PRC: perirhinal cortex, OTS: Occipital-temporal sulcus, CS: collateral sulcus.

2. The authors use the Yeo 17-Network solution for cortical organization. While this is indeed one of the most frequently used cortical parcellations in the field of human neuroimaging, the authors might want to consider a recently updated 15-network solution which was based on individualized, precision analyses (Du et al., 2024 *Journal of Neurophysiology*).

In any event, while using the 17-Network solution is totally valid, the network labels that the authors use are unclear to me. For example, to the best of my knowledge, Yeo 17-Network solution defines three DMN – DMN-A, DMN-B and DMN-C. It is unclear where does DMN-D mentioned by the authors come from. I believe the cortical network labeled as DMN-D in the manuscript is the temporal-parietal network in Yeo 17-Network solution, which was later recognized as the language network.

We thank the Reviewer for the opportunity to clarify the term “default mode network-D” (DMN-D) in our manuscript. Our naming scheme was adopted following a series of publications by Yeo and colleagues (Yeo et al., 2011; Buckner et al., 2013; Baker et al., 2014; Shinn et al., 2015). Within this naming convention, DMN-D corresponds to the **N14 label** in Yeo-17’s network, which is also referred to (in other papers from the same and from other groups) as the *temporal-parietal network* (Yeo et al., 2015; Tang et al., 2020; Kahali et al., 2021) or the *auditory/language network* (Shinn et al., 2015). Of note, this network (“DMN-D”, also known as the temporal-parietal/auditory/language network) was part of the original “DMN” in the Yeo-7 parcellation (Yeo et al., 2011; Buckner et al., 2013). Thus, we opted to maintain the “DMN-D” terminology, given the origin of the network as well as to maintain a more straightforward and consistent organization of our results in Figure 5 (shown below for your convenience). However, following the Reviewer’s excellent comment (which made us realize that there is indeed a good deal of discrepancy across the naming conventions for this same network!), we have now added a footnote to the manuscript when DMN-D is first mentioned clarifying that DMN-D is also known as the *temporo-parietal network* and the *auditory/language network* to prevent confusion.

Figure 5. EC and aHPC time-dependent pattern changes were driven by specific networks.

(A and B) The Yeo-17 networks are shown on a surface template (color-coded by network). The primary networks that drove time-related resting connectivity pattern changes in the **(A)** EC and **(B)** aHPC are highlighted. **(C-D)** Bar plots show temporal drift scores for **(C)** EC and Yeo-17 large-scale networks (Female: left, Male: right) and **(D)** aHPC and Yeo-17 large-scale networks (Female: left, Male: right). *Denotes significantly different temporal drift scores compared to the

control somatomotor network in each subject. Abbreviations: default mode network (DMN), dorsal attention network (DA), ventral attention network (VAN), somatomotor network (SMotor).

Page 18:

We found that EC time-related connectivity pattern changes were primarily driven by the DMN-C (Female: $r = -0.206$, $p_{FDR} < 0.001$; Male: $r = -0.252$, $p_{FDR} < 0.001$) and DMN-D² (Female: $r = -0.175$, $p_{FDR} = 0.002$; Male: $r = -0.193$, $p_{FDR} < 0.001$) as well as dorsal attention network A (DA-A)...

2. The DMN-D is also known as the temporal-parietal network (Yeo et al., 2015; Tang et al., 2020; Kahali et al., 2021) or the auditory/language network (Shinn et al., 2015).

Finally, while the authors use limbic networks A and B, I believe the recent paper by Girn et al., 2024, Imaging Neuroscience might be of interest. This study assigns regions of the limbic networks to DMN. I believe these recent data can help the authors to interpret some of their findings. Also please note that the updated 15-network solution (Du et al., 2024 Journal of neurophysiology) has no limbic networks.

We thank the Reviewer for pointing us to these recently-published manuscripts.

As noted by the Reviewer, the updated 15-network parcellation method (Girn et al., 2024) merges the limbic network into the default mode networks. First, it is important to note that the limbic networks did not drive large-scale resting pattern changes when EC or anterior HPC were used as seeds in our study. Moreover, given the cytoarchitectonic differences between regions of the limbic network and DMN, it is possible that adopting the 15-network parcellation method may reduce anatomical specificity of the network masks. Given this, in addition to the widespread use of the Yeo-17 network atlas (with over 9,030 citations as of April 25, 2025), we have opted to retain our original approach. This ensures that our findings maintain both anatomical specificity and broad generalizability across readers and research settings.

Nonetheless, inspired by the Reviewer's comment, we note that Du et al. (2024) raises an important point regarding the identification of the supra-areal association megacluster (SAAM) at the temporo-parietal junction (TPJ) that is relevant to our findings. The SAAM

has been proposed as a critical hub where hippocampal-predominant projections intermix with other anatomical pathways, potentially facilitating communication of the EC and HPC with broader brain networks. Consistent with this framework, our findings suggest that the temporal drifts observed in EC- and anterior HPC- resting connectivity patterns are driven by cortical networks that include the core members of SAAM (dorsal attention, default mode, and temporal-parietal networks). We have now incorporated this point into our discussion.

Page 25-26:

Recent research suggests that the DA and DMN networks juxtapose near the TPJ—and that these networks serve as a hub for EC and HPC communication with other large-scale networks (Du et al., 2024).

3. The rationale for the study presented in the introduction is currently missing important prior work addressing the functional neuroanatomy of human MTL and its cortical interactions. (1) Interactions between ERC and HIPP are important since ERC is the main gateway of cortical input and output of HIPP (Witter et al., 1989, Progress in Neurobiology). (2) Cortical interactions between ERC and HIPP are important due to the network-level analysis performed by the authors; presenting this previous work (mentioned below) will make a stronger case for the performed analyses and connect the introduction better with the results and discussion. Previous studies associated subregions of the human ERC with different regions of CA1/subiculum and even more relevant to the current study – different regions of ERC with HIPP long-axis; furthermore, anterior and posterior HIPP were found to be associated with different cortical networks, similar to different regions of ERC that associate with different cortical networks/systems (Angeli et al., 2023 bioRxiv; Zheng et al., 2021 PNAS; Reznik et al., 2024 Current Biology; Reznik et al., 2023 Neuron; Maass et al., eLife 2015; Navarro Schröder et al., 2015 eLife). These and other studies (see below) should be mentioned as the authors build the rationale for the interactions between ERC and HIPP (including anterior vs posterior HIPP) and the interactions between ERC-HIPP and distributed cortical regions.

We appreciate the Reviewer's suggestion to strengthen our study rationale by incorporating prior work on the functional neuroanatomy of EC and HPC, including the specific anatomical connections along the hippocampal longitudinal axis and their cortical network interactions. Following the Reviewer's thoughtful input, we incorporated the

following sections into our introduction, which we agree further strengthens and contextualizes our prediction and approach:

Page 3:

Prior studies suggest that the dynamics of time and temporal context—including temporal intervals, event duration, and the temporal order of events—are reflected in changes in neural activity patterns in the hippocampus (HPC) and its primary source of cortical projections, the entorhinal cortex (EC) (Lavenex and Amaral, 2000; Eichenbaum, 2017a; Radvansky and Zacks, 2017; Buzsáki and Tingley, 2018; Clewett et al., 2019; Sugar and Moser, 2019; Bellmund et al., 2020; Palombo and Cocquyt, 2020; Wang et al., 2022).

Page 4:

The HPC can be divided into anterior and posterior hippocampus (corresponding to ventral and dorsal hippocampus in the rodent) (Poppenk et al., 2013; Strange et al., 2014) which are preferentially connected to the medial and lateral subregions of the EC, respectively (Witter and Groenewegen, 1984; Witter and Amaral, 1991, 2021). Extant data suggest that the strength of tracking of time in humans may be differentially supported by anterior *versus* posterior HPC (Nielson et al., 2015) as well as by lateral *versus* medial subregions of the EC (Tsao et al., 2018; Bellmund et al., 2019; Montchal et al., 2019).

Pages 5:

Relatedly, evidence for task-evoked temporal context representations has typically been stronger in the human anterior-lateral division of EC (alEC, analogous to the lateral division in rodents) than in the posterior-medial division (pmEC, analogous to the medial division in rodents) (Maass et al., 2015; Eichenbaum, 2017b; Tsao et al., 2018; Bellmund et al., 2019; Montchal et al., 2019). Collectively, these results obtained in task contexts suggest that coding of temporal contexts at

longer timescales may be differentially supported across distinct regions of the hippocampus—i.e, in aHPC *versus* pHPC—and EC—in aEC *versus* pmEC. However, whether the strength of spontaneous coding of time in humans varies systematically across EC subregions or along the HPC longitudinal axis—putatively reflecting an anterior-posterior gradient of temporal context encoding—has not been previously tested.

Previous tracing (Witter and Groenewegen, 1984; Witter and Amaral, 1991, 2021; Naber et al., 2001) and functional connectivity studies (Zheng et al., 2021; Reznik et al., 2024; Angeli et al., 2025) indicate that the subregions of the EC and HPC also show differential inter-regional and cortical connectivity profiles. For instance, the aHPC and medial EC exhibit preferential connectivity with the default mode network (DMN), whereas the pHPC and lateral EC are strongly connected with the ventral attention network (VAN/salience network) (Maass et al., 2015; Zheng et al., 2021; Reznik et al., 2023, 2024; Angeli et al., 2025). Given the previously identified critical role of the EC and HPC function in temporal-context related representations (Eichenbaum, 2017b; Sugar and Moser, 2019), it is possible that their system-level interactions may also be sensitive to the passage of time.

4. While the authors acknowledge the division of the HIPP into anterior and posterior HIPP, the ERC is analyzed and presented as one region without intrinsic divisions. While I understand the challenge in performing detailed MR imaging of human ERC, the authors should mention that ERC divisions in the rat, lateral ERC and medial ERC, are implicated in time coding (Eichenbaum, 2017 Neuron; Tsao et al., 2018 Nature). Recent investigations into the human ERC suggest that different divisions of human ERC are associated with different cortical networks pointing to different functional roles in different subregions of human ERC. Similar to my previous comment, I believe addressing the functional and anatomical complexity of the ERC in humans and animals will build a stronger case for the rationale of the paper and the analyses performed by the authors.

We thank the Reviewer for the excellent suggestion to more directly acknowledge the functional and anatomical complexity of the EC as part of the rationale for the current

manuscript. Following the Reviewer's input, we (a) investigated differences in temporal drift between EC subregions and (b) expanded the introduction and discussion to better contextualize our predictions and results within the known functional heterogeneity of EC.

First, to investigate whether time-related resting connectivity drift differs across subregions of the EC, we segmented the EC into anterolateral (aIEC) and posteromedial (pmEC) subregions using the anatomical segmentation approach from Maass et al. (2015), which is based on anatomical landmarks (e.g., uncus, hippocampal head, etc.) and has been widely adopted in prior work (Maass et al., 2015; Olsen et al., 2017; Montchal et al., 2019; Yeung et al., 2021). Of note, these EC subregions are associated with distinct medial temporal lobe regions and cortical networks—aIEC is preferentially linked with perirhinal cortex and anterior-temporal cortical networks, while the pmEC is associated with the parahippocampal cortex and posterior-medial cortical networks (Maass et al., 2015; Navarro Schröder et al., 2015; Reznik et al., 2023, 2024). Functionally, these subregions also differ in temporal coding, with aIEC reliably representing temporal context in prior studies, whereas the pmEC typically does not (Tsao et al., 2018; Bellmund et al., 2019; Montchal et al., 2019).

Our results revealed that time-related drifts in aIEC-whole brain resting-state functional connectivity were consistently stronger than in pmEC, as reflected by (1) more negative temporal drift scores in aIEC compared to pmEC across both subjects; (2) more negative averaged voxel-wise temporal drift scores in aIEC relative to pmEC (Figure 4; see additional results below and in the main manuscript; “*Stronger time-related changes in aIEC than pmEC resting connectivity patterns*”). This finding is consistent with prior work suggesting that aIEC (functional corresponding to the lateral EC in rodents) exhibits stronger temporal coding than pmEC (Tsao et al., 2018; Bellmund et al., 2019; Montchal et al., 2019).

Figure 4. Temporal drift strength differs between EC subregions. (A, D) In both subjects, the similarity of aIEC-whole-brain resting connectivity patterns decreases with time interval elapsed between sessions, an association not as reliable (across both subjects) in pmEC. **(B, E)** Voxel-wise temporal drift scores in EC for the female **(B)** and male **(E)** subjects. **(C, F)** Averaged temporal drift scores across aIEC and pmEC voxels, demonstrating a stronger negative association with time in aIEC compared to pmEC for both female **(C)** and male **(F)** subjects.

We now added this new EC subregion analysis to our revised manuscript:

Page 2 (Abstract):

Hippocampal connectivity temporal drifts followed an anterior-to-posterior gradient, and anterolateral-EC showed stronger temporal drift than posteromedial EC.

Page 5-6 (Introduction), previously included in response to this Reviewer's Comment #3 and pasted here for your convenience:

Relatedly, evidence for task-evoked temporal context representations has typically been stronger in the human anterior-lateral division of EC (aIEC, analogous to the lateral division in rodents)

than in the posterior-medial division (pmEC, analogous to the medial division in rodents) (Maass et al., 2015; Eichenbaum, 2017b; Tsao et al., 2018; Bellmund et al., 2019; Montchal et al., 2019). Collectively, these results obtained in task contexts suggest that coding of temporal contexts at longer timescales may be differentially supported across distinct regions of the hippocampus—i.e, in aHPC *versus* pHPC—and EC—in aIEC *versus* pmEC. However, whether the strength of spontaneous coding of time in humans varies systematically across EC subregions or along the HPC longitudinal axis—putatively reflecting an anterior-posterior gradient of temporal context encoding—has not been previously tested.

Previous tracing (Witter and Groenewegen, 1984; Witter and Amaral, 1991, 2021; Naber et al., 2001) and functional connectivity studies (Zheng et al., 2021; Reznik et al., 2024; Angeli et al., 2025) indicate that the subregions of the EC and HPC also show differential inter-regional and cortical connectivity profiles. For instance, the aHPC and medial EC exhibit preferential connectivity with the default mode network (DMN), whereas the pHPC and lateral EC are strongly connected with the ventral attention network (VAN/salience network) (Maass et al., 2015; Zheng et al., 2021; Reznik et al., 2023, 2024; Angeli et al., 2025). Given the previously identified critical role of the EC and HPC function in temporal-context related representations (Eichenbaum, 2017b; Sugar and Moser, 2019), it is possible that their system-level interactions may also be sensitive to the passage of time.

Page 12-13 (Results):

Stronger time-correlated changes in aIEC than pmEC resting connectivity patterns

EC is a complex structure that can be subdivided into the aIEC and pmEC, which are associated with functionally distinct cortical networks (Reznik et al., 2023, 2024) and have been differentially linked to temporal coding (Tsao et al., 2018; Montchal et al., 2019). Here, we found that only aIEC-whole-brain resting pattern similarity reliably correlated with the time interval elapsed

between session pairs (Female: $r = -0.218$, $p < 0.001$; Male: -0.179 , $p < 0.001$), whereas the pmEC did not show a reliable association across both subjects (only in the Male subject: $r = -0.111$, $p = 0.001$; Female: $r = -0.034$, $p = 0.478$; **Figure 4 A & D, Supplementary Table 1**). Critically, the temporal drift score was significantly more negative in aIEC compared to pmEC in both subjects (Female: $z = -3.138$, $p = 0.001$; Male: $z = -1.802$, $p = 0.036$).

To further examine the potential functional divergence in temporal coding between the aIEC and pmEC, we next calculated voxel-wise temporal drift scores and averaged them per subregion. We found that the averaged time-related drift in aIEC-whole brain resting-state functional connectivity was consistently stronger than that of the pmEC, as reflected by numerically more negative temporal drift scores in the aIEC across both subjects (**Figure 4B-C and E-F**). Collectively, these findings indicate that spontaneous changes in aIEC-whole brain connectivity robustly correlate with objective time changes.

Page 15:

We found that the EC, aHPC, and aIEC whole-brain temporal drift scores were stronger than both control regions (i.e., significantly more negative; test of the difference between dependent correlation coefficients: Female EC vs. controls $ps < 0.016$; Female aHPC vs. control ROIs $ps < 0.043$; Male EC vs. control ROIs $ps < 0.004$; Male aHPC vs. control ROIs $ps < 0.05$, Female aIEC vs. control ROIs $ps < 0.006$; Male aIEC vs. control ROIs $ps < 0.027$; **Supplementary Table 1**). Of note, this was not the case for the whole HPC (which showed significantly stronger drifts than the control ROIs in the Female subject only; the difference from control ROIs in the Male subject was at trend level) or pHPC and pmEC ROIs (which did not differ significantly from Control ROIs in either subject).

Page 20 (Discussion):

Likewise, time-related drifts were stronger in aIEC than pmEC.

Page 23-24 (Discussion):

Time-related whole brain connectivity pattern changes are stronger in aIEC than pmEC

Recent electrophysiological studies in rodents (Tsao et al., 2018) and fMRI studies in humans (Bellmund et al., 2019; Montchal et al., 2019) have shown that the aIEC reliably represents temporal context, whereas the pmEC does so less consistently. For instance, neurons in the rat lateral EC exhibit firing rates that robustly encode temporal information across timescales from seconds to hours in rodents, a phenomenon less pronounced in the medial EC (Tsao et al., 2018). Similarly, increased BOLD activity in the human aIEC—but not in pmEC—was associated with greater accuracy in retrieving temporal information about when an event occurred (Montchal et al., 2019). In line with these findings, we found that the aIEC-whole brain resting connectivity patterns exhibited a stronger and more reliable temporal drift across subjects compared to pmEC, suggesting that spontaneous changes in aIEC-whole brain connectivity pattern may reflect the passage of time.

Page 30-31 (Methods):

The EC was segmented using a well-validated parcellation strategy that uses anatomical landmarks (Maass et al., 2015; Navarro Schröder et al., 2015). Briefly, the most anterior level of the EC was fully covered by the aIEC. We began delineating the pmEC at the very medial/dorsal tip of the EC, 2 mm after the appearance of the hippocampal head. The border between the aIEC and pmEC gradually shifted laterally, forming an oblique boundary relative to the medial wall. Finally, we labeled all EC voxels as pmEC where the uncus was no longer present (Maass et al., 2015; Navarro Schröder et al., 2015; Olsen et al., 2017; Yeung et al., 2021).

HPC longitudinal axis and EC subregional analysis: Temporal drift scores for the analysis of the HPC longitudinal axis was calculated as described above, but using single-voxels in the hippocampus as seeds, instead of the average timeseries in the ROI. Similarly, voxel-wise temporal drift scores in the EC were calculated and then averaged within each EC subregion.

Supplementary Table 1: Temporal drift scores (Pearson's r) and comparison of correlation coefficients

	Female				Male			
	r	p	vs. M1, p	vs. PRC, p	r	p	vs. M1, p	vs. PRC, p
EC	-0.206	< 0.001*	0.010*	0.016*	-0.217	< 0.001*	0.004*	0.001*
aHPC	-0.180	< 0.001*	0.034*	0.043*	-0.153	< 0.001*	0.051~	0.046*
HPC	-0.187	< 0.001*	0.026*	0.025*	-0.146	< 0.001*	0.093~	0.076~
pHPC	-0.142	0.002*	0.098~	0.151	-0.123	< 0.001*	0.221	0.216
alEC	-0.218	< 0.001*	0.006*	0.006*	-0.179	< 0.001*	0.027*	0.017*
pmEC	-0.034	0.227	0.592	0.808	-0.111	0.001*	0.314	0.319
M1	-0.050	0.158	-	-	-0.087	0.006*	-	-
PRC	-0.089	0.032*	-	-	-0.091	0.007*	-	-

r : Temporal drift score calculated as Pearson's correlation coefficient; p : Permutation test p -value; vs. Control, p : Comparison of dependent correlation coefficients between EC and HPC ROI vs. control regions (M1 and PRC). * Denotes permutation test $p \leq 0.05$; ~ Denotes numeric trend of permutation test $p \leq 0.1$. Bolded text denotes temporal drift scores that are statistically significant (nonparametric permutation test) and specific (relative to the control regions, as indexed by a formal comparison of dependent correlation coefficients).

5. To continue with my previous comment on a more fine-grain examination of ERC subregions, I suggest the authors to perform the voxel-wise analysis on the ERC as well (the same analysis the authors performed on the HIPPO) and to examine the difference in temporal coding along the anterior-posterior ERC axis - reflecting potential human homologues of rodent lateral ERC and medial ERC (Maass et al., eLife 2015; Navarro Schröder et al., 2015 eLife) and along the medial-lateral ERC axis, reflecting the recently reported ERC functional bands (Reznik et al., 2024, Current Biology). I believe this is an important analysis to add to the manuscript in order to examine in more detail the functional properties of human ERC and to strengthen the authors' conclusions.

We appreciate the Reviewer's thoughtful suggestion for the voxel-wise EC analysis. As shown above (this Reviewer's Comment #4), we found that the time-related drift in aIEC-whole brain resting connectivity patterns was stronger than in pmEC, consistent with prior findings across species (Tsao et al., 2018; Bellmund et al., 2019; Montchal et al., 2019).

Next, we followed the Reviewer's suggestion to probe whether there was any hint of a gradient or differentiation across the EC X-axis using a voxel-wise approach. Specifically, we examined whether there was a gradient of temporal drift scores in the medial-lateral (M-L) axis of the EC using the analogous approach for the HPC. To that end, we computed temporal drift scores along the entorhinal axes (M-L) on a voxel-wise basis and correlated each entorhinal voxel's temporal drift score with its X-axis coordinate to investigate whether there was a linear association between the magnitude of time-dependent drifts and the entorhinal axis.

Consistent with prior findings on EC functional subdivisions, we found that temporal drift scores were negatively correlated with the M-L axis coordinates (**Figure R3 A**; Pearson's correlation $r = -0.348$, $p < 0.001$), a robust association that was significantly above chance in the male subject (**Figure R3 B**; nonparametric permutation test [X-axis coordinate shuffle; $n = 5000$): $p < 0.001$]. Specifically, a stronger time-related drift was found in the lateral entorhinal voxels' resting pattern relative to medial entorhinal voxels. Notably, these findings align with recent work suggesting distinct functional bands within the EC (Reznik et al., 2024) and the role of the lateral EC in temporal representations (Eichenbaum, 2017b; Tsao et al., 2018; Bellmund et al., 2019; Montchal et al., 2019). However, note that we did not find a statistically significant gradient along M-L axis in female subject [M-L axis: Pearson's correlation $r = -0.005$; $p = 0.94$, nonparametric permutation test (x axis coordinate shuffle; $n = 5000$): $p = 0.475$, **Figure R3 C-D**].

Figure R3: Voxel-wise temporal drift scores and non-parametric permutation tests in the female (A-B) and male subject (C-D). **(A, C)** Pearson's correlation between voxel-wise temporal drift scores and the corresponding axis coordinates. **(B, D)** Nonparametric permutations tested whether the real temporal drift score (blue vertical line) was significantly above chance.

We suspect that these inconsistent findings across subjects may be due in part to the oblique orientation of the aIEC-pmEC boundary (**Figure R4**). As shown in Figure R4, the oblique border of EC does not linearly follow X-axis alone, making it challenging to derive an X coordinate axis that perfectly aligns with the aIEC-pmEC transition. Given these limitations, we opted not to include this analysis in our manuscript, but we provide them here for completeness.

Figure R4: 3D reconstruction of medial temporal lobe areas, adopted from Maass et al., 2015.

6. In page 20 the authors write that “Given that [[part of the]] ERC is part of the DMN-C” and refer to Yeo et al., 2011. To the best of my knowledge, Yeo et al do not report any associations between ERC and DMN. In fact, robust evidence for the association between ERC and different subdivisions of the canonical DMN (DMN-A and DMN-B) was provided only recently (Reznik et al., 2024, *Current Biology*; Reznik et al., 2023 *Neuron*).

We appreciate the Reviewer’s insight and the opportunity to clarify the relationship between the EC and default mode network-C (DMN-C). The EC partially overlaps with the DMN-C network as defined by Yeo-17 network atlas [Figure R5, (Yeo et al., 2011)].

Figure R5: The EC mask (black outline, Fischl et al., 2009) is shown overlaid on the default mode network C (**DMN-C**, red, (Yeo et al., 2011)). The overlap (pink) between the EC and DMN-C is shown in pink and highlighted in the inset.

We suspect that the potential confusion may stem from the existing variation in nomenclature across different studies. Specifically, following this Reviewer’s comment, we revisited the literature to compare DMN subdivision definitions by different authors, and noticed that **DMN-A** and **DMN-B** in Reznik’s et al., work (Braga and Buckner, 2017; Reznik et al., 2023, 2024); **Figure R6A**) correspond to **DMN-C** and **DMN-A** of the Yeo-17 atlas masks, respectively (Yeo et al., 2011, 2015; Baker et al., 2014; Shinn et al., 2015);

https://github.com/ThomasYeoLab/CBIG/tree/master/stable_projects/brain_parcellation/Yeo2011_fcMRI_clustering; **Figure R6B**). Under this framework, our findings of strong time-related **EC-“DMN-C”** (Yeo-17 definition) connectivity pattern change align very well with Reznik et al.’s (2023, 2024) findings of an **EC-“DMN-A”** (Braga/Reznik definition) association.

Figure R6: Definitions of the default mode network (DMN) across different publications are shown. **(A)** The DMN masks derived from multi-session hierarchical Bayesian modeling from (Reznik et al., 2024) are shown. DN-A, default network A; DN-B, default network B; FPN-A, frontoparietal network A; FPN-B, frontoparietal network B; PMN/SAL, parietal memory network/salience network. **(B)** The DMN masks from the Yeo-17 atlas from **Yeo et al., 2011** are shown. DN-A, default network A; DN-B, default network B, DN-C, default network C; DN-D/TempPar, default network D/Temporal parietal network.

We realize that the inconsistent ‘letter’->DMN subnetwork assignment across studies may introduce confusion (e.g., *DMN-MTL* in Andrews-Hanna et al. 2014; *DMN-A* in Braga and Buckner, 2017; Reznik et al., 2023, 2024). To address this, we now directly reference the terminology proposed by Andrews-Hanna et al. (2014), referring to the DMN-C as the “DMN-medial temporal lobe subsystem” (DMN-MTL) when first introducing DMN-C in our manuscript. We hope that this revision emphasizes the anatomical location of the DMN-C subnetwork as used in our manuscript (also called DMN-A in (Braga and Buckner, 2017; Reznik et al., 2023, 2024), and minimizes confusion. We include the revised manuscript text clarifying our terminology for DMN-C below.

This portion of the DMN—comprising the medial temporal lobe cortex, retrosplenial cortex, posterior cingulate cortex, and posterior portion of inferior parietal cortex—is often termed the ‘posterior-medial network’ (Andrews-Hanna et al., 2010; Ranganath and Ritchey, 2012), and corresponds to the DMN-C¹ of a widely used network parcellation (i.e, Yeo-17 network) (Buckner et al., 2008; Yeo et al., 2011; Ritchey and Cooper, 2020; Menon, 2023).

1. The DMN-C in the current study is also known as DMN-medial temporal lobe subsystem (DMN-MTL) (Andrews-Hanna et al., 2014) and DMN-A in Braga/Reznik studies (Braga and Buckner, 2017; Reznik et al., 2023).

7. Can the authors please plot the whole-brain correlations calculated using the ERC, HIPPO, M1 and PRC seeds in a supplementary figure? I believe having these data displayed can be very helpful for examining “model-free” whole-brain associations of the MTL regions and to compare them to previous reports (Libby et al., 2012 Journal of Neuroscience; Kahn et al., 2008 Journal of neurophysiology and other studies mentioned earlier).

We thank the Reviewer for this suggestion; we now have included each of these figures (for each subject) as supplementary figures (and have uploaded each of them to NeuroVault):

Page 36:

The averaged functional connectivity maps for each ROI are shared in NeuroVault (<https://neurovault.org/collections/VEHAFBWA/>, Supplementary Figure 6-7).

Supplementary Figure 6: Whole-brain functional connectivity maps for each ROI in the female subject.

Supplementary Figure 7: Whole-brain functional connectivity maps for each ROI in the male subject.

8. *I think the authors can add another exciting analysis to the manuscript. While I leave this recommendation to the discretion of the authors, I believe this analysis can provide important insights into the time tracking mechanism. Since the male subject was scanned twice a day for 10 days, it opens the opportunity to examine the similarities between within-day vs between-days sessions to examine the role of sleep in spontaneous time tracking (Marshall and Born, 2007 Trends in cognitive sciences).*

We thank the Reviewer for this suggestion. In response, we examined whether sleep influences regional-whole brain resting functional connectivity pattern changes. To do so, we analyzed session pairs separated by a 12-hour interval and created separate subsets for 'within-day' vs. 'between-day' session pairs (where within-day pairs are 12-h apart *without* intermittent sleep and between-day pairs are 12-h apart while including a night of sleep). We asked whether there were differences in functional connectivity patterns similarity between session pair subsets for each brain region of interest (EC, aHPC, pHPC, HPC, PRC, and M1). However, we did not observe statistically significant differences between within-day and between-day comparisons (all $ps > 0.164$).

These null results may reflect limitations in our experimental design, which was not optimized to detect sleep-related effects. Specifically, the number of session pairs available for this analysis was relatively small (within-day: 10 pairs, between-day: 9 pairs), potentially limiting the statistical power to detect a sleep-related effect. Thus, the current dataset may not be ideal for investigating sleep-related effects on resting state connectivity pattern changes.

9. *In page 27, the authors mention that they performed a control analysis to examine the differences in drift score between ERC/HIPP and M1/PRC, however these analyses are not directly referred to in the main text. I think these are critical comparisons and it is great that the authors performed these analyses, but I believe they should be mentioned in the main results section.*

We agree with the Reviewer that this an important control analysis that should be referred to in the main text of the Results. We had indeed previously included it as part of the Results section, titled: “*Comparing the strength of time-related drifts in EC and HPC versus control regions*”, as follows.

Page 14-15:

Next, we tested whether time-related drifts in whole-brain resting connectivity patterns are generally present when using other regions as seeds, *versus whether they are statistically*

stronger in EC- and aHPC- resting connectivity patterns compared to other regions. To that end, we compared the strength of their time correlation against two control regions: the primary motor cortex (M1) and the perirhinal cortex (PRC). The M1 was tested as a control site since contextual representations that track elapsed time are not known to be present in this region (Huber et al., 2012; Ma et al., 2020). The PRC was chosen due to its spatial and anatomical proximity to the other medial temporal lobe (MTL) regions of interest and its established role in coding for non-contextual information in the service of memory (in contrast to the EC and HPC), such as object-centered and semantic information (Hsieh et al., 2014; Zou et al., 2023).

We found that the EC, aHPC, and aIEC whole-brain temporal drift scores were stronger than both control regions (i.e., significantly more negative; test of the difference between dependent correlation coefficients: Female EC vs. controls $ps < 0.016$; Female aHPC vs. control ROIs $ps < 0.043$; Male EC vs. control ROIs $ps < 0.004$; Male aHPC vs. control ROIs $ps < 0.05$, Female aIEC vs. control ROIs $ps < 0.006$; Male aIEC vs. control ROIs $ps < 0.027$; **Supplementary Table 1**). Of note, this was not the case for the whole HPC (which showed significantly stronger drifts than the control ROIs in the Female subject only; the difference from control ROIs in the Male subject was at trend level) or pHPC and pmEC ROIs (which did not differ significantly from Control ROIs in either subject). In sum, these results suggest regional specificity in the strength of time-related EC and aHPC-whole-brain resting connectivity changes.

References:

- Albert NB, Robertson EM, Miall RC (2009) The resting human brain and motor learning. *Curr Biol* 19:1023–1027.
- Aly M, Turk-Browne NB (2016) Attention Stabilizes Representations in the Human Hippocampus. *Cereb Cortex* 26:783–796.
- Andrews-Hanna JR, Reidler JS, Sepulcre J, Poulin R, Buckner RL (2010) Functional-anatomic fractionation of the brain's default network. *Neuron* 65:550–562.
- Andrews-Hanna JR, Saxe R, Yarkoni T (2014) Contributions of episodic retrieval and mentalizing to autobiographical thought: evidence from functional neuroimaging, resting-state connectivity, and fMRI meta-analyses. *Neuroimage* 91:324–335.
- Angeli PA, DiNicola LM, Saadon-Grosman N, Eldaief MC, Buckner RL (2025) Specialization of the human hippocampal long axis revisited. *Proc Natl Acad Sci U S A* 122:e2422083122.
- Attardo A, Fitzgerald JE, Schnitzer MJ (2015) Impermanence of dendritic spines in live adult CA1 hippocampus. *Nature* 523:592–596.
- Baker JT, Holmes AJ, Masters GA, Yeo BTT, Krienen F, Buckner RL, Öngür D (2014) Disruption of cortical association networks in schizophrenia and psychotic bipolar disorder. *JAMA Psychiatry* 71:109–118.
- Banks PJ, Bashir ZI, Brown MW (2012) Recognition memory and synaptic plasticity in the perirhinal and prefrontal cortices. *Hippocampus* 22:2012–2031.
- Bellmund JLS, Deuker L, Doeller CF (2019) Mapping sequence structure in the human lateral entorhinal cortex. *Elife*:1–20.
- Bellmund JLS, Polti I, Doeller CF (2020) Sequence memory in the hippocampal-entorhinal region. *J Cogn Neurosci* 32:2056–2070.
- Berron D, Vieweg P, Hochkeppeler A, Pluta JB, Ding S-L, Maass A, Luther A, Xie L, Das SR, Wolk DA, Wolbers T, Yushkevich PA, Düzel E, Wisse LEM (2017) A protocol for manual segmentation of medial temporal lobe subregions in 7Tesla MRI. *NeuroImage: Clinical* 15:466–482.
- Bouffard NR, Golestani A, Brunec IK, Bellana B, Park JY, Barense MD, Moscovitch M (2023) Single voxel autocorrelation uncovers gradients of temporal dynamics in the hippocampus and entorhinal cortex during rest and navigation. *Cereb Cortex* 33:3265–3283.
- Braga RM, Buckner RL (2017) Parallel Interdigitated Distributed Networks within the Individual Estimated by Intrinsic Functional Connectivity. *Neuron* 95:457-471.e5.
- Brunec IK, Bellana B, Ozubko JD, Man V, Robin J, Liu ZX, Grady C, Rosenbaum RS, Winocur G, Barense MD, Moscovitch M (2018) Multiple Scales of Representation along the Hippocampal Anteroposterior Axis in Humans. *Curr Biol* 28:2129-2135.e6.

- Brunec IK, Robin J, Olsen RK, Moscovitch M, Barense MD (2020) Integration and differentiation of hippocampal memory traces. *Neurosci Biobehav Rev* 118:196–208.
- Buckner RL, Andrews-Hanna JR, Schacter DL (2008) The brain's default network: anatomy, function, and relevance to disease. *Ann N Y Acad Sci* 1124:1–38.
- Buckner RL, Krienen FM, Yeo BTT (2013) Opportunities and limitations of intrinsic functional connectivity MRI. *Nat Neurosci* 16:832–837.
- Buonomano DV, Buzsáki G, Davachi L, Nobre AC (2023) Time for Memories. *J Neurosci* 43:7565–7574.
- Buzsáki G, Tingley D (2018) Space and Time: The Hippocampus as a Sequence Generator. *Trends Cogn Sci* 22:853–869.
- Canals S, Beyerlein M, Merkle H, Logothetis NK (2009) Functional MRI evidence for LTP-induced neural network reorganization. *Curr Biol* 19:398–403.
- Chen L, Cummings KA, Mau W, Zaki Y, Dong Z, Rabinowitz S, Clem RL, Shuman T, Cai DJ (2020) The role of intrinsic excitability in the evolution of memory : Significance in memory allocation , consolidation , and updating. *Neurobiol Learn Mem* 173:107266.
- Clewett D, DuBrow S, Davachi L (2019) Transcending time in the brain : How event memories are constructed from experience. *Hippocampus*:1–22.
- Collin SHP, Milivojevic B, Doeller CF (2015) Memory hierarchies map onto the hippocampal long axis in humans. *Nat Neurosci* 18:1562–1564.
- Dandolo LC, Schwabe L (2018) Time-dependent memory transformation along the hippocampal anterior–posterior axis. *Nat Commun* 9:1–11.
- Diedenhofen B, Musch J (2015) cocor: a comprehensive solution for the statistical comparison of correlations. *PLoS One* 10:e0121945.
- Dresler M, Shirer WR, Konrad BN, Müller NCJ, Wagner IC, Fernández G, Czisch M, Greicius MD (2017) Mnemonic training reshapes brain networks to support superior memory. *Neuron* 93:1227-1235.e6.
- Driscoll LN, Duncker L, Harvey CD (2022) Representational drift: Emerging theories for continual learning and experimental future directions. *Curr Opin Neurobiol* 76:102609.
- Du J, DiNicola LM, Angeli PA, Saadon-Grosman N, Sun W, Kaiser S, Ladopoulou J, Xue A, Yeo BTT, Eldaief MC, Buckner RL (2024) Organization of the human cerebral cortex estimated within individuals: networks, global topography, and function. *J Neurophysiol* 131:1014–1082.
- Eichenbaum H (2017a) Time (and space) in the hippocampus. *Curr Opin Behav Sci* 17:65–70.
- Eichenbaum H (2017b) On the Integration of Space, Time, and Memory. *Neuron* 95:1007–1018.

- Frankó E, Insausti AM, Artacho-Pérula E, Insausti R, Chavoix C (2014) Identification of the human medial temporal lobe regions on magnetic resonance images. *Hum Brain Mapp* 35:248–256.
- Girn M, Setton R, Turner GR, Spreng RN (2024) The “limbic network,” comprising orbitofrontal and anterior temporal cortex, is part of an extended default network: Evidence from multi-echo fMRI. *Netw Neurosci* 8:860–882.
- Grotzinger H, Pritschet L, Shapurenka P, Santander T, Murata EM, Jacobs EG (2024) Diurnal fluctuations in steroid hormones tied to variation in intrinsic functional connectivity in a densely sampled male. *J Neurosci* 44:e1856232024.
- Harmelech T, Malach R (2013) Neurocognitive biases and the patterns of spontaneous correlations in the human cortex. *Trends Cogn Sci* 17:606–615.
- Harmelech T, Preminger S, Wertman E, Malach R (2013) The Day-After Effect: Long Term, Hebbian-Like Restructuring of Resting-State fMRI Patterns Induced by a Single Epoch of Cortical Activation. *J Neurosci* 33:9488–9497.
- Harris KM (1999) Structure, development, and plasticity of dendritic spines. *Curr Opin Neurobiol* 9:343–348.
- Hayashi-Takagi A, Yagishita S, Nakamura M, Shirai F, Wu YI, Loshbaugh AL, Kuhlman B, Hahn KM, Kasai H (2015) Labelling and optical erasure of synaptic memory traces in the motor cortex. *Nature* 525:333–338.
- Hsieh L-T, Gruber MJ, Jenkins LJ, Ranganath C (2014) Hippocampal Activity Patterns Carry Information about Objects in Temporal Context. *Neuron*:1165–1178.
- Huber D, Gutnisky DA, Peron S, O'Connor DH, Wiegert JS, Tian L, Oertner TG, Looger LL, Svoboda K (2012) Multiple dynamic representations in the motor cortex during sensorimotor learning. *Nature* 484:473–478.
- Insausti R, Amaral DG (2004) *The Human Nervous System*. Elsevier.
- Insausti R, Juottonen K, Soininen H, Insausti AM, Partanen K, Vainio P, Laakso MP, Pitkänen A (1998) MR volumetric analysis of the human entorhinal, perirhinal, and temporopolar cortices. *AJNR Am J Neuroradiol* 19:659–671.
- Kahali S, Raichle ME, Yablonskiy DA (2021) The role of the human brain neuron-Glia-synapse composition in forming resting-state functional connectivity networks. *Brain Sci* 11:1565.
- Keinath AT, Wang ME, Wann EG, Yuan RK, Dudman JT, Muzzio IA (2014) Precise spatial coding is preserved along the longitudinal hippocampal axis. *Hippocampus* 24:1533–1548.
- Kong R, Li J, Orban C, Sabuncu MR, Liu H, Schaefer A, Sun N, Zuo X-N, Holmes AJ, Eickhoff SB, Yeo BTT (2019) Spatial Topography of Individual-Specific Cortical Networks Predicts Human Cognition, Personality, and Emotion. *Cereb Cortex* 29:2533–2551.

- Kong R, Yang Q, Gordon E, Xue A, Yan X, Orban C, Zuo X-N, Spreng N, Ge T, Holmes A, Eickhoff S, Yeo BTT (2021) Individual-specific areal-level parcellations improve functional connectivity prediction of behavior. *Cereb Cortex* 31:4477–4500.
- Krenz V, Alink A, Sommer T, Rooszendaal B, Schwabe L (2023) Time-dependent memory transformation in hippocampus and neocortex is semantic in nature. *Nat Commun* 14:6037.
- Lavenex P, Amaral DG (2000) Hippocampal-neocortical interaction: a hierarchy of associativity. *Hippocampus* 10:420–430.
- Li Q, Wang X, Wang S, Xie Y, Li X, Xie Y, Li S (2018) Musical training induces functional and structural auditory-motor network plasticity in young adults. *Hum Brain Mapp* 39:2098–2110.
- Ma Z, Liu H, Komiyama T, Wessel R (2020) Stability of motor cortex network states during learning-associated neural reorganizations. *J Neurophysiol* 124:1327–1342.
- Maass A, Berron D, Libby LA, Ranganath C, Düzel E (2015) Functional subregions of the human entorhinal cortex. *Elife* 4:1–20.
- Magee JC, Grienberger C (2020) Synaptic plasticity forms and functions. *Annu Rev Neurosci* 43:95–117.
- Mankin EA, Diehl GW, Leutgeb S, Leutgeb JK, Mankin EA, Diehl GW, Sparks FT, Leutgeb S, Leutgeb JK (2015) Hippocampal CA2 Activity Patterns Change over Time to a Larger Extent than between Spatial Contexts. *Neuron* 85:190–201.
- Mankin EA, Sparks FT, Slayyeh B, Sutherland RJ, Leutgeb S, Leutgeb JK (2012) Neuronal code for extended time in the hippocampus. *Proc Natl Acad Sci U S A* 109:19462–19467.
- Marks TD, Goard MJ (2021) Stimulus-dependent representational drift in primary visual cortex. *Nat Commun* 12:1–16.
- Massey PV, Phythian D, Narduzzo K, Warburton EC, Brown MW, Bashir ZI (2008) Learning-specific changes in long-term depression in adult perirhinal cortex. *J Neurosci* 28:7548–7554.
- Mau W, Sullivan DW, Kinsky NR, Hasselmo ME, Howard MW, Eichenbaum H (2018) The Same Hippocampal CA1 Population Simultaneously Codes Temporal Information over Multiple Timescales. *Curr Biol* 28:1499-1508.e4.
- Menon V (2023) 20 years of the default mode network: A review and synthesis. *Neuron* 111:2469–2487.
- Montchal ME, Reagh ZM, Yassa MA (2019) precise temporal memories are supported by the lateral entorhinal cortex in humans. *Nat Neurosci* 22 Available at: <http://dx.doi.org/10.1038/s41593-018-0303-1>.

- Moyer CE, Zuo Y (2018) Cortical dendritic spine development and plasticity: insights from in vivo imaging. *Curr Opin Neurobiol* 53:76–82.
- Naber PA, Lopes Da Silva FH, Witter MP (2001) Reciprocal connections between the entorhinal cortex and hippocampal fields CA1 and the subiculum are in register with the projections from CA1 to the subiculum. *Hippocampus* 11:99–104.
- Navarro Schröder T, Haak KV, Zaragoza Jimenez NI, Beckmann CF, Doeller CF (2015) Functional topography of the human entorhinal cortex. *Elife* 4:1–17.
- Nielson DM, Smith TA, Sreekumar V, Dennis S, Sederberg PB (2015) Human hippocampus represents space and time during retrieval of real-world memories. *Proc Natl Acad Sci U S A* 112:11078–11083.
- Olsen RK, Palombo DJ, Rabin JS, Levine B, Ryan JD, Rosenbaum RS (2013) Volumetric analysis of medial temporal lobe subregions in developmental amnesia using high-resolution magnetic resonance imaging: Hippocampal Subregions In Developmental Amnesia. *Hippocampus* 23:855–860.
- Olsen RK, Yeung L-K, Noly-Gandon A, D'Angelo MC, Kacollja A, Smith VM, Ryan JD, Barense MD (2017) Human anterolateral entorhinal cortex volumes are associated with cognitive decline in aging prior to clinical diagnosis. *Neurobiol Aging* 57:195–205.
- Palombo DJ, Amaral RSC, Olsen RK, Müller DJ, Todd RM, Anderson AK, Levine B (2013) KIBRA polymorphism is associated with individual differences in hippocampal subregions: Evidence from anatomical segmentation using high-resolution MRI. *Journal of Neuroscience* 33:13088–13093.
- Palombo DJ, Cocquyt C (2020) Emotion in Context : Remembering When. *Trends Cogn Sci* 24:687–690.
- Poldrack RA et al. (2015) Long-term neural and physiological phenotyping of a single human. *Nat Commun* 6:8885.
- Poppenk J, Evensmoen HR, Moscovitch M, Nadel L (2013) Long-axis specialization of the human hippocampus. *Trends Cogn Sci* 17:230–240.
- Pritschet L, Santander T, Taylor CM, Layher E, Yu S, Miller MB, Grafton ST, Jacobs EG (2020) Functional reorganization of brain networks across the human menstrual cycle. *Neuroimage* 220:117091.
- Pruessner JC, Köhler S, Crane J, Pruessner M, Lord C, Byrne A, Kabani N, Collins DL, Evans AC (2002) Volumetry of temporopolar, perirhinal, entorhinal and parahippocampal cortex from high-resolution MR images: considering the variability of the collateral sulcus. *Cereb Cortex* 12:1342–1353.
- Radvansky GA, Zacks JM (2017) Event Boundaries in Memory and Cognition. *Curr Opin Behav Sci* 17:133–140.

- Ranganath C, Ritchey M (2012) Two cortical systems for memory-guided behaviour. *Nat Rev Neurosci* 13:713–726.
- Raut RV, Snyder AZ, Raichle ME (2020) Hierarchical dynamics as a macroscopic organizing principle of the human brain. *Proc Natl Acad Sci U S A* 117:20890–20897.
- Reddy L, Zoefel B, Possel JK, Peters J, Dijksterhuis DE, Poncet M, van Straaten ECW, Baayen JC, Idema S, Self MW (2021) Human Hippocampal Neurons Track Moments in a Sequence of Events. *J Neurosci* 41:6714–6725.
- Reznik D, Margulies DS, Witter MP, Doeller CF (2024) Evidence for convergence of distributed cortical processing in band-like functional zones in human entorhinal cortex. *Curr Biol* 34:5457-5469.e2.
- Reznik D, Trampel R, Weiskopf N, Witter MP, Doeller CF (2023) Dissociating distinct cortical networks associated with subregions of the human medial temporal lobe using precision neuroimaging. *Neuron* 111:2756-2772.e7.
- Ritchey M, Cooper RA (2020) Deconstructing the Posterior Medial Episodic Network. *Trends Cogn Sci* 24:451–465.
- Rule ME, O’Leary T, Harvey CD (2019) Causes and consequences of representational drift. *Curr Opin Neurobiol* 58:141–147.
- Sami S, Miall RC (2013) Graph network analysis of immediate motor-learning induced changes in resting state BOLD. *Front Hum Neurosci* 7:166.
- Schoonover CE, Ohashi SN, Axel R, Fink AJP (2021) Representational drift in primary olfactory cortex. *Nature* 594:541–546.
- Segal M (2017) Dendritic spines: Morphological building blocks of memory. *Neurobiol Learn Mem* 138:3–9.
- Shimbo A, Izawa E-I, Fujisawa S (2021) Scalable representation of time in the hippocampus.
- Shinn AK, Baker JT, Lewandowski KE, Öngür D, Cohen BM (2015) Aberrant cerebellar connectivity in motor and association networks in schizophrenia. *Front Hum Neurosci* 9:134.
- Strange BA, Witter MP, Lein ES, Moser EI (2014) Functional organization of the hippocampal longitudinal axis. *Nat Rev Neurosci* 15:655–669.
- Sugar J, Moser M-B (2019) Episodic memory: Neuronal codes for what, where, and when. *Hippocampus* 29:1190–1205.
- Tambini A, D’Esposito M (2020) Causal contribution of awake post-encoding processes to episodic memory consolidation. *Curr Biol* 30:3533-3543.e7.
- Tambini A, Ketz N, Davachi L (2010) Enhanced brain correlations during rest are related to memory for recent experiences. *Neuron* 65:280–290.

- Tang S, Sun N, Floris DL, Zhang X, Di Martino A, Yeo BTT (2020) Reconciling dimensional and categorical models of autism heterogeneity: A brain connectomics and behavioral study. *Biol Psychiatry* 87:1071–1082.
- Taylor CM, Pritschet L, Olsen RK, Layher E, Santander T, Grafton ST, Jacobs EG (2020) Progesterone shapes medial temporal lobe volume across the human menstrual cycle. *Neuroimage* 220:117125.
- Tompariy A, Duncan K, Davachi L (2015) Consolidation of associative and item memory is related to post-encoding functional connectivity between the ventral tegmental area and different medial temporal lobe subregions during an unrelated task. *J Neurosci* 35:7326–7331.
- Tsao A, Sugar J, Lu L, Wang C, Knierim JJ, Moser M-B, Moser EI (2018) Integrating time from experience in the lateral entorhinal cortex. *Nature* 561:57–62.
- Umbach G, Kantak P, Jacobs J, Kahana M, Pfeiffer BE, Sperling M (2020) Time cells in the human hippocampus and entorhinal cortex support episodic memory. :1–12.
- Wang J, Tambini A, Lapate RC (2022) The tie that binds: temporal coding and adaptive emotion. *Trends Cogn Sci* Available at: <http://dx.doi.org/10.1016/j.tics.2022.09.005>.
- Witter MP, Amaral DG (1991) Entorhinal Cortex of the Monkey .5. Projections to the Dentate Gyrus, Hippocampus, and Subicular Complex. *JCompNeurol* 307:437–459.
- Witter MP, Amaral DG (2021) The entorhinal cortex of the monkey: VI. Organization of projections from the hippocampus, subiculum, presubiculum, and parasubiculum. *J Comp Neurol* 529:828–852.
- Witter MP, Groenewegen HJ (1984) Laminar origin and septotemporal distribution of entorhinal and perirhinal projections to the hippocampus in the cat. *J Comp Neurol* 224:371–385.
- Xia J, Marks TD, Goard MJ, Wessel R (2021) Stable representation of a naturalistic movie emerges from episodic activity with gain variability. *Nat Commun* 12:5170.
- Xu T, Yu X, Perlik AJ, Tobin WF, Zweig JA, Tennant K, Jones T, Zuo Y (2009) Rapid formation and selective stabilization of synapses for enduring motor memories. *Nature* 462:915–919.
- Yang G, Pan F, Gan W-B (2009) Stably maintained dendritic spines are associated with lifelong memories. *Nature* 462:920–924.
- Yeo BTT, Krienen FM, Sepulcre J, Sabuncu MR, Lashkari D, Hollinshead M, Roffman JL, Smoller JW, Zöllei L, Polimeni JR, Fischl B, Liu H, Buckner RL (2011) The organization of the human cerebral cortex estimated by intrinsic functional connectivity. *J Neurophysiol* 106:1125–1165.
- Yeo BTT, Tandi J, Chee MWL (2015) Functional connectivity during rested wakefulness predicts vulnerability to sleep deprivation. *Neuroimage* 111:147–158.

- Yeung L-K, Hale C, Rizvi B, Igwe K, Sloan RP, Honig LS, Small SA, Brickman AM (2021) Anterolateral entorhinal cortex volume is associated with memory retention in clinically unimpaired older adults. *Neurobiol Aging* 98:134–145.
- Yu M, Song H, Huang J, Song Y, Liu J (2020) Motor Learning Improves the Stability of Large-Scale Brain Connectivity Pattern. *Front Hum Neurosci* 14:571733.
- Yushkevich PA et al. (2015a) Quantitative comparison of 21 protocols for labeling hippocampal subfields and parahippocampal subregions in in vivo MRI: towards a harmonized segmentation protocol. *Neuroimage* 111:526–541.
- Yushkevich PA, Piven J, Hazlett HC, Smith RG, Ho S, Gee JC, Gerig G (2006) User-guided 3D active contour segmentation of anatomical structures: significantly improved efficiency and reliability. *Neuroimage* 31:1116–1128.
- Yushkevich PA, Pluta JB, Wang H, Xie L, Ding SL, Gertje EC, Mancuso L, Klot D, Das SR, Wolk DA (2015b) Automated volumetry and regional thickness analysis of hippocampal subfields and medial temporal cortical structures in mild cognitive impairment. *Hum Brain Mapp* 36:258–287.
- Zheng A et al. (2021) Parallel hippocampal-parietal circuits for self- and goal-oriented processing. *Proc Natl Acad Sci U S A* 118 Available at: <http://dx.doi.org/10.1073/pnas.2101743118>.
- Ziv Y, Burns LD, Cocker ED, Hamel EO, Ghosh KK, Kitch LJ, El Gamal A, Schnitzer MJ (2013) Long-term dynamics of CA1 hippocampal place codes. *Nat Neurosci* 16:264–266.
- Zou F, Wanjia G, Allen EJ, Wu Y, Charest I, Naselaris T, Kay K, Kuhl BA, Hutchinson JB, DuBrow S (2023) Re-expression of CA1 and entorhinal activity patterns preserves temporal context memory at long timescales. *Nat Commun* 14:4350.
- Zvi NR, Elisha PM (2023) Representations in human primary visual cortex drift over time. *Nat Commun* 14:4422.

REVIEWERS' COMMENTS

Reviewer #1 (Remarks to the Author):

The authors have done an excellent job of addressing my concerns, and I have no further comments.

Reviewer #1 (Remarks on code availability):

I did not try to run the code but I did have a look. The authors provided code for the entire pipeline which is great. The code provides a readme file but it is a little sparse on instructions. For example, I don't see instructions on installing required software, directory structure required to run the code, variables that might need to be changed to get it to run, etc. The Readme file does contain information on the order the code should be run. The code itself is annotated which is good.

We thank Reviewer 1 for this suggestion; we have now carefully expanded and clarified the README file in our GitHub repository (<https://github.com/LEAPNeuroLab/rsFCTemporalDrift>).

Reviewer #2 (Remarks to the Author):

I thank the authors for thoroughly addressing all my comments. I find it very interesting that aIEC showed stronger temporal drift than pmEC. This finding aligns well with stronger temporal drift in aHPC and overall, makes the manuscript more conceptually broad. I have only a few minor suggestions:

On page 4 the authors write that "The HPC can be divided into anterior and posterior hippocampus (corresponding to ventral and dorsal hippocampus in the rodent) which are preferentially connected to the medial and lateral subregions of the EC, respectively." And then the authors add "Extant data suggest that the strength of tracking of time in humans may be differentially supported by anterior versus posterior HPC as well as by lateral versus medial subregions of the EC." Medial and lateral subregions of the EC that are differently connected to the anterior vs posterior hippocampus are not the same "lateral versus medial subregions of the EC". Therefore, I suggest that the authors write "The HPC can be divided into anterior and posterior hippocampus (corresponding to ventral and dorsal hippocampus in the rodent) which are preferentially connected to different parts of the EC".

We thank Reviewer 2 for the insightful suggestion. We have edited the manuscript accordingly.

Page 4:

The HPC can be divided into anterior and posterior hippocampus (corresponding to ventral and dorsal hippocampus in the rodent)^{35,36}, which are differentially connected to the EC³⁷⁻³⁹.

On page 15, the authors write "We found that the EC, aHPC, and aIEC ..." - aIEC is part of EC, so the phrasing is a bit odd.

We have now edited the sentence as below:

Page 11 (used to be on Page 15):

We found that the EC—particularly aIEC—and aHPC-whole-brain temporal drift scores were stronger than both control regions ...

Page 17, footnote 1 - many studies other than Braga/Reznik refer to DMN-C as DMN-A - e.g., DiNicola et al. (2020) Journal of neurophysiology; Angeli et al. (2025) PNAS; Du et al. (2024) Journal of neurophysiology. I believe that the authors can refer to the other naming convention (DMN-A) and provide the relevant references, similar to referring to the naming convention of the DMN-MTL system.

We thank Reviewer 2 for the additional citations for the DMN nomenclature. We have now included all three citations in what used to be footnote 1 and is now part of the main text on Page 13 (used to be on Page 17):

This portion of the DMN—comprising the medial temporal lobe cortex, retrosplenial cortex, posterior cingulate cortex, and posterior portion of inferior parietal cortex—is often termed the ‘posterior-medial network’^{75,76}, and corresponds to the DMN-C (also referred to as the DMN-medial temporal lobe subsystem (DMN-MTL)⁷⁷ or DMN-A^{57,58,78–80}) of a widely used network parcellation (i.e, Yeo-17 network)^{74,81–83}.

In the discussion (page 20), I believe it might be clearer to discuss first the EC role in the temporal drift and then mention EC divisions. Currently the authors mention aIEC and pmEC and then go back to EC. This might be confusing for the readers.

We thank Reviewer 2 for noting this structural issue, which made us realize that the transition between summarizing what was common *versus* distinct for subregions in EC and HPC was not as clear as it could have been. We have now edited the first paragraph of the Discussion section to first summarize the findings without regional differences, which is followed by one sentence for HPC longitudinal axis differences, and one for EC regional differences, respectively, before transitioning to summarizing network-level results.

Page 14 (used to be on Page 20):

Extant work has shown that signals in the EC and HPC reflect changes in temporal context across a variety of task demands and species^{1,3-6,8,9,33}. Here, we investigated whether intrinsic fluctuations in EC- and HPC-whole-brain functional connectivity patterns reflect the passage of time in the absence of task demands. Using a dense-sampling study, we found that **both** EC- and HPC-whole-brain resting connectivity patterns became increasingly dissimilar with longer time intervals across a 30-day period. Moreover, time-related changes in HPC showed a functional gradient along the longitudinal axis, wherein aHPC was characterized by stronger time-dependent drifts compared to pHPC. Likewise, time-related drifts were stronger in aIEC than pmEC.

I have no further comments and I congratulate the authors on this great manuscript!

We thank the Reviewer for their supportive comments!